# Efficient Planning in Large MDPs with Weak Linear Function Approximation

**Roshan Shariff**
University of Alberta & Amii
roshan.shariff@ualberta.ca

**Csaba Szepesvári**
DeepMind & University of Alberta & Amii
szepesva@ualberta.ca

## Abstract

Large-scale Markov decision processes (MDPs) require planning algorithms with runtime independent of the number of states of the MDP. We consider the planning problem in MDPs using linear value function approximation with only weak requirements: low approximation error for the optimal value function, and a small set of "core" states whose features span those of other states. In particular, we make no assumptions about the representability of policies or value functions of non-optimal policies. Our algorithm produces almost-optimal actions for any state using a generative oracle (simulator) for the MDP, while its computation time scales polynomially with the number of features, core states, and actions and the effective horizon.

## 1 Introduction

Markov decision processes (MDPs) are commonly used to model sequential decision making under uncertainty and have a wide range of applications [see 42, 33, 7, for example]. We consider planning in large-scale, expected discounted total reward MDPs. Computing an optimal policy in the discounted setting is known to require "reading" all states at least once [6]. As the state space for most interesting applications is intractably large if not infinite ("Bellman's curse of dimensionality"), it is common to consider restrictions to the problem that can allow efficient calculation of near-optimal actions. One such relaxation is *online planning* — the MDP can be accessed through a simulator and we ask only for a good action at a given state [25, 29]. While in this problem the complexity of computing a "good action" can be independent of the number of states, the complexity is exponential in the planning horizon [25]. An alternative idea, which can be traced back to at least the work of Schweitzer and Seidmann [35], is to assume that one has access to a *feature representation* (that is, a vector of features for each state) and the planner needs to work well for those MDPs where the *optimal value function* of the MDP can be uniformly well approximated over all states by some appropriate weighted combination of the features. Since an accurate approximation of the optimal value function is known to be sufficient to generate near-optimal behavior,[1] the problem simplifies to producing a good estimate of the unknown feature weights with a computation cost that is independent of the number of states.

In this paper we consider the intersection of these two problem formulations. More precisely, our goal is to construct planning algorithms that produce an action for any given input state, using black-box access to the MDP through a simulator which takes a state and an action as input and produces a random next state and immediate reward. The planner can also access the feature representation of any state as a $d$-dimensional feature vector. Assume that the optimal value function of the MDP can be uniformly well approximated — to an accuracy of $\varepsilon_{\text{approx}}$ — as a linear combination of the features with fixed, but unknown, coefficients. Our goal is a randomized planning algorithm that interacts with

the simulator $\text{poly}(1/\varepsilon, d, H, A, \dots)$ times to produce an action, such that following this action in every state results in an $O(\varepsilon + c\varepsilon_{\text{approx}})$-optimal policy. Here $H = 1/(1 - \gamma)$ is the effective planning horizon for the discount factor $0 \le \gamma < 1$, which is used in the definition of the values of policies; $A$ is the number of actions; and $c > 0$ is an error inflation factor that may depend on $\gamma$, $d$, and $A$.

We call the features "weak" as we only require the optimal value function to be accurately representable by their linear combinations, in contrast to "strong" features that can accurately represent the value functions of *all* policies; in this latter case the problem of efficient planning in the presence of a simulator is known to have a solution, both in the episodic and discounted settings [43, 28]. As pointed out by Du et al. [12], with only weak features, the problem of efficient planning has not yet been solved.

**Our Contributions**   We design a randomized algorithm that positively answers the challenge posed above under one extra assumption — that the feature vectors of all states lie within the convex hull of the feature vectors of a few selected "core states" that the algorithm is given. In particular, we show that the query-complexity and runtime of our algorithm is polynomial in the relevant quantities and the number of core states, providing a partial positive answer to the previously open problem of efficient planning in the presence of weak features.

To achieve our result, we start from the *approximate linear programming* (ALP) approach where the value function is approximated using the feature vectors. Following Lakshminarayanan et al. [27], we construct a *relaxed* ALP that drops all constraints except at the core states. In their work, Lakshminarayanan et al. gave bounds on the error of the value function that is obtained from solving this relaxed ALP. The authors also suggested a way to turn this error bound into an efficient planning method, though without a detailed analysis. The main contribution of the present work is to fill this gap, in addition to simplifying, strengthening and streamlining the earlier results. In particular, we propose using a randomized saddle-point solver that substantially reduces the computational requirements compared to the procedure hinted at by [27].

**Paper Organization**   The rest of the paper is organized as follows: Sections 1.1 and 1.2 give background on MDPs and introduce the linear programming (LP) approach to planning. Section 2 formally defines the problem. Then, in Section 3, we present the linear program that we start with and give our first results, bounding the value loss of the policy that can be read out from optimal solutions of the linear program. Section 4 gives the efficient algorithm to solve the linear program and our main result. Section 5 discusses related work. The paper is concluded in Section 6. The proofs of the results are moved to Appendix A in the Supplementary Material.

**Notation**   The set of real numbers is denoted by $\mathbb{R}$, whereas $\mathbb{R}_+ = [0, \infty)$. $\mathbb{R}^d$ denotes the vectors with $d$ dimensions, while the $m \times n$ matrices are $\mathbb{R}^{m \times n}$. We use bold letters for vectors ($\boldsymbol{r}$) and bold capitals for matrices ($\boldsymbol{P}$); their elements are written as $r_i$ and $P_{i,j}$ and matrix rows are $\boldsymbol{P}_i$. For vectors of identical dimension, $\boldsymbol{x} \le \boldsymbol{y}$ means element-wise comparison: $x_i \le y_i$ for each index $i$. The standard basis vector $\boldsymbol{e}_i$ has $e_{i,i} = 1$ and $e_{i,j} = 0$ for $i \ne j$, and the constant 0 or 1 vector is denoted by $\boldsymbol{0}, \boldsymbol{1} \in \mathbb{R}^d$; their dimension depends on the context. All vectors are considered column vectors by default. The probability simplex over any finite set $\mathcal{A}$ is denoted $\Delta_{\mathcal{A}} := \{\boldsymbol{p} \in \mathbb{R}_+^{|\mathcal{A}|} \mid \|p\|_1 = 1\}$. For a finite set $\mathcal{S}$ with cardinality $S = |\mathcal{S}|$, we will think of functions $v : \mathcal{S} \to \mathbb{R}$ or $\varphi : \mathcal{S} \to \mathbb{R}^d$ as vectors or matrices, respectively, and use both notations: for example, $\boldsymbol{v} \in \mathbb{R}^S$, $v_s$, or $v(s)$; and $\boldsymbol{\Phi} \in \mathbb{R}^{S \times d}$, $\boldsymbol{\varphi}_s$, or $\boldsymbol{\varphi}(s)$ where $s \in \mathcal{S}$. When the domain takes the form $\mathcal{S} \times \mathcal{A}$ with respective cardinalities $S$ and $A$, we use intuitive double indices of the form $sa$, e.g., with $r : \mathcal{S} \times \mathcal{A} \to \mathbb{R}$ we index components of $\boldsymbol{r}$ using the notation $r_{sa}$ (i.e., $r_{sa} = r(s, a)$). In this case we also write $\boldsymbol{r} \in \mathbb{R}^{SA}$.

For convenience, an Index of Notation section is included in the Supplementary Material.

## 1.1   Background

A (finite, discounted) Markov Decision Process (MDP) is defined by the entities $(\mathcal{S}, \mathcal{A}, \boldsymbol{P}, \boldsymbol{r}, \gamma)$ where $\mathcal{S}$ and $\mathcal{A}$ are finite sets of states and actions, respectively. Without loss of generality we let $\mathcal{S} = [S]$ and $\mathcal{A} = [A]$, using the notation $[n] = \{1, \dots, n\}$ for integers $n > 0$. When the process is in state $s \in \mathcal{S}$ and action $a \in \mathcal{A}$ is chosen, a random reward is received with expectation $r_{sa} \in \mathbb{R}$ and the process transitions to a new state $s' \in \mathcal{S}$ with probability $P_{sa,s'}$. For convenience, we arrange the transition probabilities into a matrix $\boldsymbol{P} \in \mathbb{R}^{SA \times S}$ and the rewards into a vector $\boldsymbol{r} \in \mathbb{R}^{SA}$. Thus, $\boldsymbol{P}$ is a

*row-stochastic matrix* — each row $P_{sa}$ for a state $s$ and action $a$ is a valid probability distribution for the next state.

For our purposes it will be sufficient to consider stationary policies, which we will just call *policies*. A policy $\pi : S \to \Delta_{\mathcal{A}}$ is a function from states to probability distributions over actions — we use $\pi(a|s)$ to denote the probability assigned by $\pi$ to action $a$ in state $s$. Following a policy means that upon visiting state $s$, an action $a \sim \pi(s)$ is chosen at random. This gives rise to an infinite sequence of states, actions, and corresponding rewards. The *value* $v_\pi(s)$ of a policy for a process starting at state $s \in S$ is defined as the total expected $\gamma$-discounted sum of the rewards incurred:

$$v_\pi = \sum_{t=0}^{\infty} (\gamma P_\pi)^t r_\pi, \quad \text{where } [r_\pi]_s = \sum_{a \in \mathcal{A}} \pi(a|s)\, r_{sa}, \quad \text{and } [P_\pi]_{s,s'} = \sum_{a \in \mathcal{A}} \pi(a|s)\, P_{sa,s'}; \quad (1)$$

$r_\pi \in \mathbb{R}^S$ is the expected reward and $P_\pi \in \mathbb{R}^{S \times S}$ is the state transition matrix. A policy $\pi^*$ is called *optimal* if $v_{\pi^*} \geq v_\pi$ for every policy $\pi$, where the inequality is component-wise. Every MDP has an optimal policy, and all optimal policies have the same value function $v^*$, the *optimal value function*. Further, there always exist *deterministic* optimal policies, which concentrate all their probability on a single action for each state. We will also need $q^* \in \mathbb{R}^{SA}$, which is defined via $q^* = r + \gamma P v^*$.

## 1.2 Linear Programming

Our approach to the MDP planning problem is based on the standard linear programming (LP) formulation; for details, see Puterman [32, §6.9], who swaps the primal and dual problems:

$$v^*(s_0) = \min \left\{ e_{s_0}^\mathsf{T} v \;\middle|\; v \in \mathbb{R}^S, \qquad r + (\gamma P - E)v \leq 0 \right\} \qquad \text{(Primal LP)}$$

$$= \max \left\{ \mu^\mathsf{T} r \;\middle|\; \mu \in \mathbb{R}_+^{SA}, \;\; e_{s_0} + \mu^\mathsf{T}(\gamma P - E) = 0 \right\}. \qquad \text{(Dual LP)}$$

The matrix $E : \mathbb{R}^{SA \times S}$ has elements $E_{sa,s} = 1$ and $E_{sa,s'} = 0$ for $s \neq s'$. It maps vectors from $\mathbb{R}^S$ to $\mathbb{R}^{SA}$ by duplicating their elements over all actions: $[Ev]_{sa} = v_s$ for all $s, a$. Both the primal and dual optimization problems above have the same optimal value: the optimal value of state $s_0$. The dual variables $\mu$ are *discounted state-action occupancy measures* of a policy $\pi$ starting at state $s_0$:

$$\mu^\mathsf{T} = \rho_\pi^\mathsf{T} \sum_{t=0}^{\infty} (\gamma \tilde{P}_\pi)^t, \quad \text{where } [\rho_\pi]_{sa} = \pi(a|s_0)\, e_{s,s_0}, \quad \text{and } [\tilde{P}_\pi]_{sa,s'a'} = \pi(a'|s')[P_\pi]_{sa,s'}; \quad (2)$$

$\rho_\pi \in \Delta_{S \times \mathcal{A}}$ is the initial distribution over state-action pairs and $\tilde{P}_\pi \in \mathbb{R}^{SA \times SA}$ is the state-action transition probability matrix; the state transition matrix $P_\pi$ is defined in (1). The constraint $e_{s_0} + \mu^\mathsf{T}(\gamma P - E) = 0$ enforces that $\mu$ has this form, and is therefore generated by some stochastic policy. Thus the dual problem can be seen as a linear formulation of policy optimization, where policies are represented by their occupancy measures and the objective is to maximize expected discounted reward — in particular, the policy corresponding to any feasible $\mu$ can be obtained by conditioning on the state: $\pi_\mu(a|s) = \mu_{sa}/\sum_{a'} \mu_{sa'}$ for any state with non-zero occupancy measure.

The *approximate linear program* (ALP) of Schweitzer and Seidmann [35] is obtained from (Primal LP) by restricting $v$ to lie in the span of a *feature matrix* $\Phi \in \mathbb{R}^{S \times d}$ — $v = \Phi\theta$ for $\theta \in \mathbb{R}^d$. This reduces the number of variables from $S$ to the feature dimension $d$, but is still intractable to solve because of the many constraints — one for each state-action pair. The *relaxed ALP* of Lakshminarayanan et al. [27] addresses this issue by keeping only a small number of constraints that are positive linear combinations of the original constraints; this is the foundation of our approach.

## 2 Problem Definition

In the *online MDP planning problem*, a randomized planner is given a state of the MDP $s_0 \in S$ as input, and needs to return an action [e.g., 25, 16]. Letting $\pi(a|s_0)$ denote the probability that action $a$ is returned for input $s_0$, the planner's *value loss* at state $s$ is defined as $v^*(s) - v_\pi(s)$. The goal is to design planning algorithms with a small value loss for *every* state $s$ regardless of the MDP.

For large MDPs, we want the computation time to be independent of the number of states $S$ and depend polynomially on the number of actions and the "planning horizon" $H = 1/(1 - \gamma)$. To make this even remotely possible, we assume that the planner has access to a suitable feature map $\varphi : S \to \mathbb{R}^d$ (Assumption 1 below), is given a suitable set of "core states" (Assumption 2), and can

access a simulator of the MDP (Assumption 3).[2] Further, the planner is only required to perform well if the optimal value function lies uniformly close to the span of the features, which means that

$$\varepsilon_{\text{approx}} := \inf_{\boldsymbol{\theta} \in \mathbb{R}^d} \max_{s \in \mathcal{S}} |v^*(s) - \boldsymbol{\varphi}_s^\mathsf{T} \boldsymbol{\theta}| \tag{3}$$

is small. To be precise, the planner's value loss is allowed to degrade with $\varepsilon_{\text{approx}}$. Note that the class of MDPs where $\varepsilon_{\text{approx}}$ is small for a given feature map is a strict superset of those which are nearly linear up to the error $\varepsilon_{\text{approx}}$ in the sense of Jin et al. [19]. Hence, we call the features *weak* because we only require that $\varepsilon_{\text{approx}}$ as defined in (3) be small. Du et al. [12] posed the open problem of designing efficient online planning algorithms under this condition.

**Assumptions.** For the convenience of the reader, we now restate our assumptions in a concise form:

1. *Features:* The planner can access $\boldsymbol{\varphi}(s) \in \mathbb{R}^d$ for any state $s \in \mathcal{S}$. Further, there is some $\boldsymbol{\eta} \in \mathbb{R}^d$ such that $\boldsymbol{\varphi}_s^\mathsf{T} \boldsymbol{\eta} = 1$ for all $s$ — this can be ensured easily by adding a "bias" feature that is always 1.

2. *Core States:* There is a set of core states $\mathcal{S}_* \subset \mathcal{S}$ (with $|\mathcal{S}_*| = m$) that are available to the algorithm, and the feature vector of every other state can be written as a positive linear combination of the core state features: $\boldsymbol{\Phi} = \mathbf{Z}\boldsymbol{\Phi}_*$ for some non-negative matrix $\mathbf{Z} \in \mathbb{R}_+^{S \times m}$, where $\boldsymbol{\Phi} \in \mathbb{R}^{S \times d}$ and $\boldsymbol{\Phi}_* \in \mathbb{R}^{m \times d}$ consist of the stacked feature vectors for all states and the core states, respectively. Note that $\mathbf{Z}$ need not be known to the planning algorithm.

3. *Simulator:* The planner can call a randomized function $\text{SIMULATE}(s, a)$ that returns $s' \sim \mathbf{P}_{sa}$ and a reward $\hat{r}$ with $\mathbb{E}[\hat{r}] = r_{sa}$. For simplicity, we assume $|\hat{r}| \le 1$.

We do not need to explicitly assume that the feature vectors of all states lie within the convex hull of the core state features — that is a consequence of Assumptions 1 and 2 (specifically, $\mathbf{1} \in \text{span } \boldsymbol{\Phi}$).[3] Under Assumption 2, each of the core states is a "soft state aggregation" [36] that respects the feature representation. Zanette et al. [45] impose a similar requirement for core states, observing that "anchoring" the value function at states with "extreme" feature representations (i.e., on the boundary of the convex hull of feature vectors) results in the values of other states being *interpolated* (not *extrapolated*) from the values of the core states using their respective feature representations — this is sufficient to accurately deduce the values of all states when $\varepsilon_{\text{approx}}$ is small [43].

Without assuming extra structure, finding a set of core states requires checking the feature vectors of all states, which is intractable in large MDPs. It remains an interesting question what extra structure would make it possible to discover near-minimal core sets with an effort independent of the size of the MDP. We also note that for some feature maps the size of the minimal core set can be as large as the number of states $S$. Since the run time of our algorithm depends on the size of the core set, one should obviously avoid such feature maps. It remains an intriguing question whether requiring a small core set is necessary for efficient planning.

## 3 CoreLP — A Linear Program for Planning with Core States

Throughout this section, we will use $s_0 \in \mathcal{S}$ to refer to the current planning state — our goal is to output a random close-to-optimal action $a \sim \pi(s_0)$. Consider the following result, which follows immediately from the well-known "performance difference lemma" [23, Lemma 6.1]:

**Proposition 1.** Let $\pi$ be an arbitrary policy. Then,

$$\max_{s \in \mathcal{S}} v^*(s) - v_\pi(s) \le \frac{1}{1 - \gamma} \max_{s_0 \in \mathcal{S}} \mathbb{E}_{a \sim \pi(s_0)}[v^*(s_0) - q^*(s_0, a)].$$

In light of this, we will design a randomized planning algorithm that guarantees $\mathbb{E}_{a \sim \pi(s_0)}[q^*(s_0, a)] \approx v^*(s_0)$ for any input state $s_0$. Our approach stems from the relaxed linear program of Lakshminarayanan et al. [27] — more precisely, we start with the ALP but keep only the constraints corresponding to the actions of the core states and the current planning state (see Section 1.2). We then construct the corresponding dual LP, to which we add another constraint that allows us to "read out" an action distribution from the values of the dual variables.

Recall that $\mathcal{S}_* = \{s_1, \ldots, s_m\}$ is the set of core states and define $\mathcal{S}_+$ as the sequence $(s_0, s_1, \ldots, s_m)$. Note that $s_0$ is always the first state in $\mathcal{S}_+$ but may appear again if it is also a core state. For each of the $1 + m$ states in $\mathcal{S}_+$, we will select the $A$ constraints in the ALP corresponding to the state-action pairs $(s_i, a) \in \mathcal{S}_+ \times \mathcal{A}$ — a total of $(1 + m)A$ constraints. Unlike the ALP, our linear program CoreLP is based on the (Dual LP) of Section 1.2; the name refers to the core states and features that, respectively, CONSTRAIN and RELAX it.

**Theorem 2** (CoreLP)**.** *Suppose Assumptions 1 and 2 hold, $s_0 \in \mathcal{S}$, $\boldsymbol{\varphi}_0 \coloneqq \boldsymbol{\varphi}_{s_0}$, $W \in \{0, 1\}^{(1+m)A \times SA}$ has rows $[W_{s_i a}]_{s_i \in \mathcal{S}_+, a \in \mathcal{A}} = \boldsymbol{e}_{s_i a}$, and $\Lambda \coloneqq \{\boldsymbol{\lambda} \in \mathbb{R}_+^{(1+m)A} \mid \sum_{a \in \mathcal{A}} \lambda_{s_0 a} = 1\}$. Define*

$$V^\dagger = \max \left\{ \boldsymbol{\lambda}^\mathsf{T} W \boldsymbol{r} \;\middle|\; \boldsymbol{\lambda} \in \Lambda, \; \boldsymbol{\varphi}_0^\mathsf{T} + \boldsymbol{\lambda}^\mathsf{T} W (\gamma P - E) \boldsymbol{\Phi} = \boldsymbol{0} \right\}. \tag{CoreLP}$$

*Let $\boldsymbol{\lambda}^\dagger \in \Lambda$ be a maximizer of (CoreLP) and let $\boldsymbol{\pi}^\dagger \in \Delta_\mathcal{A}$ be given by $\pi^\dagger(a) = \lambda^\dagger_{s_0 a}$. Then*

$$|V^\dagger - v^*(s_0)| \le \frac{10\gamma\varepsilon_{\text{approx}}}{1 - \gamma}, \qquad v^*(s_0) - \sum_{a \in \mathcal{A}} \pi^\dagger(a) \, q^*(s_0, a) \le \frac{20\gamma\varepsilon_{\text{approx}}}{1 - \gamma}.$$

This bound matches up to constant factors (and improves by a $\gamma$ factor) the landmark result of de Farias and Van Roy [14] for the approximation error of the ALP (defined in Section 1.2). In other words, core states satisfying Assumption 2 lead to essentially no additional error in the solution of (CoreLP) compared to the ALP — this was pointed out by Lakshminarayanan et al. [27], whose result we improve upon in Theorem 5 (Appendix A.1). The theorem also implies that the linear program is both bounded and feasible, meaning its value is not $\pm\infty$; this is an important consideration when relaxing the ALP [4]. We present the detailed proof in Appendix A.2 of the Supplementary Material.

The feature matrix $\boldsymbol{\Phi}$, which in the ALP constrains the value functions of (Primal LP), instead *relaxes* the constraint in (Dual LP) to be $[\boldsymbol{e}_{s_0}^\mathsf{T} + \boldsymbol{\mu}^\mathsf{T}(\gamma P - E)]\boldsymbol{\Phi} = \boldsymbol{0}$. As a result, $\boldsymbol{\mu}$ may no longer be a discounted state-action occupancy distribution, although it behaves like one with respect to expectations of functions in the span of $\boldsymbol{\Phi}$ — using the notation of Sections 1.1 and 1.2, any feasible $\boldsymbol{\mu}$ satisfies $\boldsymbol{\mu}^\mathsf{T} E \boldsymbol{f} = \boldsymbol{e}_{s_0}^\mathsf{T} \sum_{t=0}^\infty (\gamma P_\pi)^t \boldsymbol{f}$ for some policy $\pi$ and any $\boldsymbol{f} = \boldsymbol{\Phi}\boldsymbol{\theta}$; compare this with (2).

Conversely, the $W$ matrix *constrains* $\boldsymbol{\mu}$ to be non-zero only on the actions of core states and the current planning state: $\boldsymbol{\mu} = \boldsymbol{\lambda}^\mathsf{T} W$. The discounted visits to all other states are "soft-aggregated" as positive linear combinations of the core states, as discussed in Section 2. Such aggregation is acceptable because *(i)* the above relaxation means $\boldsymbol{\mu}$ only needs to be accurate for functions in the span of $\boldsymbol{\Phi}$; and *(ii)* Assumption 2 ensures that the features of all states lie in the convex hull of the core state features. Thus the simultaneous constraint and relaxation complement each other, incurring the same $O(\varepsilon_{\text{approx}}/(1 - \gamma))$ error as the ALP which also restricts value functions to the span of $\boldsymbol{\Phi}$.

Significantly, this theorem also specifies how to select an action that achieves the promised value for the planning state $s_0$. This is made possible by *(i)* adding $s_0$ to the set of core states; and *(ii)* requiring (via the definition of $\Lambda$) that $\mu_{s_0 a} \equiv \lambda_{s_0 a}$ sum to one. This last constraint forces $\boldsymbol{\mu}$ to directly represent the action probabilities at the planning state, not indirectly by being aggregated as linear combinations of the core state actions. As a result, a solution $\boldsymbol{\mu} \equiv \boldsymbol{\lambda}^\mathsf{T} W$ of (CoreLP) yields an almost-optimal action distribution $\pi(a|s_0) = \mu_{s_0 a}$ for the planning state $s_0$.

Unfortunately, the soft state aggregation which makes (CoreLP) tractable to solve (as in Section 4) comes at a price — $\boldsymbol{\mu}$ directly encodes only an action distribution for the current planning state, not a policy for other states (unlike the original (Dual LP) of Section 1.2). We believe that solving a separate optimization problem for each planning state is unavoidable without restricting ourselves to a compactly representable class of policies; see Section 5 for a discussion of such approaches.

As a final remark, the value loss of the policy resulting from Theorem 2 is $O(\gamma\varepsilon_{\text{approx}}/(1 - \gamma)^2)$. Here, an extra $1/(1 - \gamma)$ factor is incurred in Proposition 1, while the other $1/(1 - \gamma)$ factor is incurred in Theorem 2. This is similar to the bounds obtained in previous works [e.g., 14, 43, 28].

## 4  CoreStoMP — A Stochastic Saddle-Point Algorithm

Having formulated the planning problem as a linear program with few variables and constraints, the remaining issue is that the constraints still involve quantities of the form $WP\boldsymbol{\Phi}$, which cannot be calculated exactly in time independent of $S$. However, since these are actually expectations, the simulator can be used to estimate them. One possibility would be to estimate $\boldsymbol{P}_{sa}\boldsymbol{\Phi} \in \mathbb{R}^d$ for the initial

and core states and use a plug-in estimator — often called *sample average approximation*. Instead, we pursue the *stochastic approximation* approach — using well-known first-order optimization methods to directly solve (CoreLP) by using the simulator to produce stochastic estimates of gradients that are intractable to compute exactly [20, 21]. This optimization-based approach is attractive to us because the resulting algorithm, by design, is *incremental* and *anytime* — the quality of the solution steadily improves if the algorithm is given more time.

We first rewrite (CoreLP) as an unconstrained "saddle point" problem, retaining only the constraint introduced by the definition of $\Lambda$:

$$V^\dagger = \max_{\lambda \in \Lambda} \min_{\theta \in \mathbb{R}^d} \left[ f(\lambda, \theta) := \lambda^\mathsf{T} W r + \varphi_0^\mathsf{T} \theta + \lambda^\mathsf{T} W(\gamma P - E)\Phi\theta \right]. \qquad \text{(Saddle CoreLP)}$$

To be able to use first-order methods, we calculate the gradients of $f$:

$$f_\lambda(\theta) := \nabla_\lambda f(\lambda, \theta) = W(r + (\gamma P - E)\Phi\theta), \qquad (4)$$

$$f_\theta(\lambda) := \nabla_\theta f(\lambda, \theta) = \varphi_0^\mathsf{T} + \lambda^\mathsf{T} W(\gamma P - E)\Phi. \qquad (5)$$

Note that $f$ is *bilinear:* its gradient with respect to $\theta$ only depends linearly on $\lambda$, and vice versa. The transition probabilities, which present the major computational challenge, appear only through the matrix $B := W(\gamma P - E)\Phi$, whose rows correspond to state-action pairs $(s, a) \in \mathcal{S}_+ \times \mathcal{A}$. Each row $B_{sa} = \gamma P_{sa}\Phi - \varphi_s^\mathsf{T}$ is the (discounted) expected change in the feature vector when taking action $a$ in state $s$, which suggests how to estimate it using the simulator — define $\Delta\varphi(s, s') := \gamma\varphi_{s'} - \varphi_s$ and sample $s' \sim P_{sa}$; then $\Delta\tilde{\varphi} := \Delta\varphi(s, s')$ is an unbiased estimator of $B_{sa}$. The construction of the matrix $W$ ensures that $s$ is either the current state or one of the core states. Further, we only use $s'$ through its feature representation $\varphi_{s'}$. Putting all this together, our gradient estimates are:

$$[\hat{f}_\lambda(\theta)]_{sa} := \hat{r} + \Delta\varphi(s, s')^\mathsf{T}\theta, \qquad \forall s \in \mathcal{S}_+, a \in \mathcal{A}, \quad \text{where } (\hat{r}, s') \sim \text{SIMULATE}(s, a), \qquad (6)$$

$$\hat{f}_\theta(\lambda) := \varphi_0^\mathsf{T} + \|\lambda\|_1 \Delta\varphi(s, s'), \qquad \text{where } (s, a) \sim \lambda/\|\lambda\|_1 \text{ and } s' \sim \text{SIMULATE}(s, a). \qquad (7)$$

Sampling both gradients requires a total of $1 + (1 + m)A$ queries of the simulator and an additional $O(dmA)$ computation time. By slightly abusing notation, we will use $\xi \sim \hat{f}_\theta(\lambda)$ (and $\rho \sim \hat{f}_\lambda(\theta)$) to denote a random $d$-dimensional (resp., $(1 + m)A$-dimensional) vector taken from the distribution of $\hat{f}_\theta(\lambda)$ (resp., that of $\hat{f}_\lambda(\theta)$) as defined above. Finally, we remark in passing that the gradient estimate $[\hat{f}_\lambda(\theta)]_{sa}$ is the "temporal difference error" [38] of the value function $\Phi\theta$ at state $s$ with action $a$.

We use these gradient estimates with the Stochastic Mirror-Prox algorithm of Juditsky et al. [22]. Instantiating the algorithm requires several choices — for the dual variables $\lambda \in \mathbb{R}_+^{(1+m)A}$, we use the 1-norm and the "unnormalized negentropy" regularizer; for the primal variables $\theta \in \mathbb{R}^d$ we use the norm $\|\theta\| = \|\Phi_*\theta\|_2$ and the regularizer $\|\theta\|^2/2$. The result is Algorithm 1 (CoreStoMP).

**Theorem 3** (CoreStoMP). *Suppose Assumptions 1, 2, and 3 hold, and define*

$$B := \frac{(9/8)\sqrt{m}}{1 - \gamma}, \qquad C := \frac{(9/4)\sqrt{m(1 + 2\log A + 2\gamma\log m)}}{(1 - \gamma)^2}.$$

*Let $\hat{\lambda}$ be the result of running Algorithm 1 for $T$ iterations with the parameter $B$ and the step size $\eta = C^{-1}\sqrt{2/7T}$, which requires $2T(1 + (1 + m)A)$ simulator queries. Define $\hat{\pi} \in \Delta_\mathcal{A}$ by $\hat{\pi}(a) = \hat{\lambda}_{s_0 a}$ (as in Theorem 2) and $a \sim \hat{\pi}$. Then*

$$v^*(s_0) - \mathbb{E}[q^*(s_0, a)] \le \frac{32\varepsilon_{\text{approx}}}{1 - \gamma} + \frac{21}{2(1 - \gamma)^2}\sqrt{\frac{3m(1 + 2\log A + 2\gamma\log m)}{T}}.$$

Note that the expectation on the left-hand side is both for the randomness of the algorithm and the action $a$. While the bound does not have a direct dependence on the dimension of the features, the number of core states, $m$, must exceed the rank of $\Phi$. It is notable that the approximation error does not get inflated by a rank-related quantity, as one would expect in the worst-case [28]; this is due to Assumption 2. The increase in the leading term of the approximation error compared to Theorem 2 is because of the need to bound the domain of $\theta$ by $B$; it remains for future work to avoid this necessity. Altogether, Algorithm 1 gives the following positive result for the online planning problem for MDPs.

**Corollary 4.** *Under Assumptions 1, 2, and 3, Algorithm 1 is a randomized planning algorithm that, for any $\varepsilon > 0$, uses $O(m^2 A(1 + \log A + \gamma\log m)/\varepsilon^2)$ simulator queries and $\text{poly}(d, A, m, 1/\varepsilon)$ computation to output an action. Following this action in every state gives a stochastic policy with value loss at most $O(\varepsilon_{\text{approx}}/(1 - \gamma)^2 + \varepsilon/(1 - \gamma)^3)$.*

---

**Algorithm 1** CoreStoMP: Stochastic Mirror-Prox for Planning with Core States

---

**Parameters:** $T, B, \eta$

**Initialization:** $\theta_0 \leftarrow \mathbf{0} \in \mathbb{R}^d$, $[\lambda_0]_{s_0 a} \leftarrow 1/A$, $[\lambda_0]_{sa} \leftarrow \gamma/((1-\gamma)mA)$     $\forall s \in \mathcal{S}_*, a \in \mathcal{A}$

**for** $\tau = 1, 2, \ldots, T$ **do**

    $(\theta'_\tau, \lambda'_\tau) \leftarrow \text{ProxUpdate}(B, \eta, (\theta_{\tau-1}, \lambda_{\tau-1}), (\xi, \rho))$     where $\xi \sim \hat{f}_\theta(\lambda_{\tau-1})$, $\rho \sim \hat{f}_\lambda(\theta_{\tau-1})$

    $(\theta_\tau, \lambda_\tau) \leftarrow \text{ProxUpdate}(B, \eta, (\theta_{\tau-1}, \lambda_{\tau-1}), (\xi', \rho'))$   where $\xi' \sim \hat{f}_\theta(\lambda'_\tau)$, $\rho' \sim \hat{f}_\lambda(\theta'_\tau)$

**end for**

**return** $\left(\Sigma_{\tau=1}^T \lambda_\tau\right)/T$

---

**function** ProxUpdate$(B, \eta, (\theta, \lambda), (\xi, \rho))$

    $\tilde{\theta}$   $\leftarrow \theta - \eta\xi$

    $\theta'$   $\leftarrow \tilde{\theta}/\max\{1, \|\mathbf{\Phi}_*\theta\|_2/B\}$

    $\tilde{\lambda}$   $\leftarrow \exp(\log \lambda + \eta\rho)$

    $\lambda'_{s_0} \leftarrow \tilde{\lambda}_{s_0}/\|\tilde{\lambda}_{s_0}\|_1$             where $\tilde{\lambda}_{s_0} := [\tilde{\lambda}_{s_0 a}]_{a \in \mathcal{A}}$ and similarly for $\lambda'$.

    $\lambda'_* \leftarrow (\gamma/(1-\gamma))\tilde{\lambda}_*/\|\tilde{\lambda}_*\|_1$     where $\tilde{\lambda}_* := [\tilde{\lambda}_{sa}]_{s \in \mathcal{S}_*, a \in \mathcal{A}}$ and similarly for $\lambda'$.

    **return** $(\theta', \lambda')$

**end function**

---

## 5 Related Work

The online MDP planning problem formulation we adopt — where the planner is given an input state and asked to produce a close-to-optimal action using a generative model of the MDP as a subroutine — was proposed by Kearns et al. [25] as an alternative to requiring a compact, structured representation of the MDP. Their approach, also adopted by Kocsis and Szepesvári [26] for their UCT algorithm, is to build a (sparse) look-ahead tree. Generally, the problem is that the tree needs to be sufficiently deep and the branching factor can be as large as the number of actions, which leads to an exponential blow-up as a function of the planning horizon (see footnote 2 on page 4). The focus is thus to characterize those MDPs where the planning time can be kept polynomial in the effective horizon [30, 16].

**Planning with Feature Representations**     The broader context of this work is the problem posed by the recent paper of Du et al. [12], which asks whether "good features" (or representation) are sufficient in various RL contexts — including efficient online planning in large MDPs with a generative model. Their main (negative) result states that even when the features are good enough to represent the action-value functions of all policies up to a uniform error of $\varepsilon_{\text{approx}}$, a planning algorithm that is required to produce an $O(\varepsilon_{\text{approx}})$-optimal policy needs to check at least $2^H$ states in some $H$-horizon episodic problems. Lattimore et al. [28] along with Van Roy and Dong [40] point out that if the feature space is $d$-dimensional, the exponential blowup with the planning horizon can be avoided if the policy only needs to be $O(\varepsilon_{\text{approx}} \sqrt{d} H^2)$-optimal (where the horizon is $H = 1/(1-\gamma)$, as their results are for discounted problems). They also describe an instance of approximate policy iteration that achieves this bound with $\tilde{O}(d/(\varepsilon_{\text{approx}}^2(1-\gamma)^4))$ queries, where $\tilde{O}$ hides logarithmic factors.

For the finite-horizon setting, Du et al. also present a positive result [12, Theorem C.1] for the case when a simulator of the environment is available and the optimal action-value function can be represented with no error (i.e., $\varepsilon_{\text{approx}} = 0$). The proposed method is a randomized algorithm — an instance of fitted value iteration. In addition to the usual inputs, the algorithm also takes as input $\delta$, a target failure probability. The algorithm returns an optimal policy with probability $1 - \delta$, while issuing at most poly$(d, H, \log(1/\delta), 1/\rho)$ queries to the simulator, where $\rho$ is the minimum action-value gap that also needs to be known to the algorithm. The algorithm also relies on an oracle to construct a "core set" of $d$ state-action pairs for each stage of the $H$-horizon problem whose feature vectors form a *barycentric spanner* of the set of all feature vectors at that stage. The idea of the algorithm is to construct a policy backwards by estimating the action value functions via interpolation: In each stage, the action-value of each member of the core set is estimated by using sufficiently many rollouts

using the policy constructed for the further stages. The estimated values are used with barycentric interpolation to produce values for all the other state-action pairs.

For the same finite-horizon setting but allowing for an $\varepsilon_{\text{approx}}$ error in approximating the optimal action-value function, Zanette et al. [45] describe a similar algorithm. The main difference is that their algorithm uses the estimated values in a Monte Carlo procedure in place of policy roll-outs. They also propose using a core set (which they call the anchor points) and a similar barycentric extrapolation procedure. Unfortunately, the errors propagate multiplicatively between the stages and thus, in the worst case, the error can be as large as $C^H$ where $C > 1$ depends on the choice of the features. Lattimore et al. [28] show that $1 \leq C \leq \sqrt{d}$; we note in passing that "state aggregation" gives rise to $C = 1$.

A number of authors have studied the problem of learning and planning with exact linear optimal action-value function under various extra conditions. Positive results have been shown for deterministic MDPs [41], the so-called "low Bellman rank" MDPs [18], and under a specific low variance and large gap condition [13]. Yang and Wang [43] assume the transition matrix has a linear structure and also use least-squares regression with data from a pre-selected collection of anchor state/action pairs. Their assumption — the same as ours — is that the features of all state-action pairs can be written as convex combinations of the anchoring features. They show that their algorithm needs at most $\text{poly}(d, 1/(1 - \gamma), \log(1/\delta), m)$ queries, where $m$ is the number of anchor points. Their bound scales linearly with $H^7$ where $H = 1/(1 - \gamma)$. Their result also applies to the "misspecified" case when the linear structure is only true up to a fixed error. In contrast to these results, we do not assume that the transition matrix has special structure; we make the weaker assumption that the optimal value function lies close to the span of the features.

**Approximate Linear Programming**   The narrower context of the present work is the so-called approximate linear programming (ALP) approach to approximate planning in large MDPs, described in Section 1.2. The seminal work of de Farias and Van Roy [14] showed that the ALP solution's error, compared to the optimal value function, is within a constant factor (involving $1/(1 - \gamma)$) of the best approximation error achievable by linear combinations of the given features. Unfortunately, as discussed earlier, the ALP has too many constraints to be tractable for large MDPs. Most subsequent work is therefore aimed at designing methods that keep the approximation guarantees without having to enumerate all the constraints. Schuurmans and Patrascu [34] and Guestrin et al. [17] propose using "constraint generation" for problems with additional structure (i.e., factorized transitions), while de Farias and Van Roy [15] propose randomly generating a subset of constraints from some a priori fixed distribution. All these methods require computation time that depends on uncontrolled quantities, such as the so-called induced width of a cost-network [17], or the discrepancy between the sampling distribution and the (unknown) optimal stationary distribution [15]. The fundamental difficulty is that when too many constraints are dropped, the linear program may become unbounded. To protect against this, de Farias and Van Roy [15] add an extra constraint on the optimization variables, but their bound then degrades to the *worst* approximation error over this constraint set.

Petrik and Zilberstein [31] demonstrate that the $1/(1 - \gamma)$ blow-up of the error in the bound of de Farias and Van Roy [14] can be tight. They also propose techniques to avoid it — one of them is to add extra constraints induced by short action sequences; another is to replace the hard constraints in the ALP with smooth ones with an associated Lagrange multiplier. Desai et al. [11] propose a specific way to choose the Lagrange multiplier, for which they also obtain error bounds and demonstrate improved empirical behavior. However, as they build on the work of de Farias and Van Roy [15], their results inherit the limitations of this latter work: the large number of constraints. Bhat et al. [5] extend the work of Desai et al. [11] to nonparametric function approximation. Lakshminarayanan et al. [27] depart from constraint sampling and consider the error induced by linearly combining constraints. Petrik and Zilberstein [31], in addition to the above mentioned contributions, also give error bounds for the ALP obtained by replacing the transition matrix with a sample-average estimate.

**The Dual Linear Program**   A parallel line of research aims to solve (an approximation of) the (Dual LP) optimization problem, in contrast to the aforementioned work focusing on (Primal LP). Recall from Section 1.2 that the dual variables $\boldsymbol{\mu}$ are occupancy distributions over state-action pairs generated by policies — a common theme in these approaches is to approximate such distributions using low-dimensional "distribution features". Abbasi-Yadkori et al. [1, 2] propose a stochastic gradient descent to be used on the Lagrangian derived from the dual LP and derive a policy

suboptimality bound for the resulting poly-time algorithm; however, their results only apply under some restrictive conditions.

A major advantage of the (Dual LP) is that its solutions directly encode optimal policies (as discussed in Section 1.2) rather than just value functions. When the dual variables are approximated using "distribution features", however, only a restricted class of policies can be represented. For example, when the distribution features are the occupancy measures of a given set of "base policies", then solving the approximate dual LP means finding the best mixture of the base policies. Banijamali et al. [3] present an algorithm for this problem — under the additional assumption that the occupancy measures of the base policies have large overlap. They also show that, in general, this problem is NP-hard to even approximate — finding the best stochastic policy in a restricted class can be *harder* than finding an optimal policy of the MDP. We note in passing that the assumption of a restricted class of policies that contains a close-to-optimal policy can be considered complementary to our setting, where the optimal value function is close to the span of a given feature representation.

**Primal-Dual Methods**   There has been significant interest in applying recent advances in primal-dual online optimization methods to planning in MDPs. Since the (Primal LP) optimizes value functions while the (Dual LP) optimizes occupancy measures (i.e. policies, indirectly), primal-dual optimization can be seen as an "actor-critic" approach that finds both policies and value functions simultaneously. Cogill [10] proposes solving the saddle-point form of the LP in Section 1.2, with no approximation and assuming full knowledge of the transition matrix. Chen and Wang [9] adopt the same approach but with stochastic updates using random samples of state transitions. Chen et al. [8] extend this idea to large MDPs using low-dimensional feature representations to approximate both the primal and dual variables. Bas-Serrano and Neu [4] identify a "coherence" condition on the primal and dual feature representations that is necessary to extract close-to-optimal policies from saddle-point solutions with function approximation — they also point out that, without such an assumption, the policy suboptimality bound of Chen et al. can scale with the number of states of the MDP (or worse). We can avoid this issue in our setting because we use the approximate solutions of (Saddle CoreLP) only to select one action, not an entire policy, unlike all these cited works.

# 6   Conclusions

We presented an approach to efficient online planning in large-scale $\gamma$-discounted MDPs in the presence of *(i)* a (relatively) weak $d$-dimensional feature representation; *(ii)* a core set of $m$ states whose features' convex hull covers the features of other states; and *(iii)* a stochastic simulator of the MDP. Our main contribution is an online planning algorithm that, for any target precision $\varepsilon$, achieves a value loss of $O(\varepsilon_{\text{approx}}/(1-\gamma)^2 + \varepsilon)$, where $\varepsilon_{\text{approx}}$ (3) is the best achievable error in uniformly approximating the optimal value function of the MDP using the given feature representation. When the MDP has $A$ actions per state, the algorithm's runtime is $\text{poly}(1/\varepsilon, d, m, A, 1/(1-\gamma))$, which is independent of the number of states in the MDP. Our work builds upon the approximation error bound of Lakshminarayanan et al. [27] for relaxations of the approximate linear program.

Du et al. [12] point out that it remains an open problem whether query-efficient planning is possible in large MDPs using only a simulator and features that have a small approximation error $\varepsilon_{\text{approx}}$, with no additional assumptions. Our algorithm resolves this open problem in the special case when a small set of core states is available, i.e., when $m = \text{poly}(d)$. It remains an intriguing question whether this assumption can be removed without jeopardizing efficient planning. Other interesting questions are whether the results can be extended to smoothed ALPs [11], and whether the adaptive constraint generation of Petrik and Zilberstein [31] can be used to reduce the dependence on the planning horizon.

To achieve our results, we make several novel technical contributions: We slightly change the ALP approach of Lakshminarayanan et al. [27], adding extra constraints and using a saddle-point formulation. We then show that near-optimal action distributions can be extracted from approximate solutions of the saddle-point problem. We solve the saddle-point problem using a stochastic approximation algorithm, Stochastic Mirror-Prox [22] — a first-order primal-dual optimization method that uses stochastic gradient estimates, which in our case are provided by the simulator. We believe that these techniques and ideas can find applications in other problems beyond our work.

## Broader Impact

Our research has the nature of basic science — we are working on foundational improvements to reinforcement learning algorithms. We are not targeting any specific applications, and it is hard to foresee any societal consequences beyond those brought about by advancing the state of our knowledge of machine learning.

## Acknowledgements

Csaba Szepesvári gratefully acknowledges funding from the Canada CIFAR AI Chairs Program, the Alberta Machine Intelligence Institute (Amii), and the Natural Sciences and Engineering Research Council of Canada (NSERC).

## Footnotes

[1]See Proposition 1; or, for example, Szepesvári [39, Lemma 5.17], Kearns et al. [25, Lemma 5], Kallenberg [24, Theorem 3.7].

[2]With no additional assumptions on the MDP, any online planning algorithm implementing an $\varepsilon$-suboptimal policy may need up to $\Omega((1/\varepsilon)^{H-1})$ simulator queries to find each action; this is exponential in the planning horizon $H$ for any constant $\varepsilon > 0$ — see Kearns et al. [25, Theorem 2], noting that their $H$ is different from ours.

[3]We have $\mathbf{Z}\mathbf{1} = \mathbf{Z}\boldsymbol{\Phi}_*\boldsymbol{\eta} = \boldsymbol{\Phi}\boldsymbol{\eta} = \mathbf{1}$ for some $\boldsymbol{\eta} \in \mathbb{R}^d$, so the rows of $\mathbf{Z}$ must sum to one.

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
