[Supplementary Material]

# Supplementary Material

## Index of Notation

For the convenience of the reader, we have collected the most frequently used symbols and their meanings in the following table:

| | |
|---|---|
| $\mathbb{R}, \mathbb{R}_+$ | real numbers; non-negative real numbers. |
| $\mathbb{R}^d, \mathbb{R}^{m \times n}$ | $d$-dimensional vectors; matrices of size $m \times n$. |
| $e_i, \mathbf{0}, \mathbf{1}$ | standard basis vector: $e_{i,i} = 1$ and $e_{i,j} = 0$ for $i \neq j$; constant zero or one vectors. |
| $a \oplus b$ | concatenation of vectors: if $a \in \mathbb{R}^n$ and $b \in \mathbb{R}^m$, then $a \oplus b \in \mathbb{R}^{n+m}$. |
| $\mathcal{S}, \mathcal{A}, \mathcal{S}_*$ | sets of states, actions, and core states (Section 1.1 and Assumption 2). |
| $S, A, m$ | number of states $|\mathcal{S}|$, actions $|\mathcal{A}|$, and core states $|\mathcal{S}_*|$, respectively. |
| $s_0, s, s', a$ | planning state (Section 3) and other states $\in \mathcal{S}$; actions $\in \mathcal{A}$. |
| $P, E$ | row-stochastic matrices in $\mathbb{R}_+^{SA \times S}$ (Sections 1.1 and 1.2); $E_{sa} = e_s \in \mathbb{R}^S$. |
| $r, \hat{r}$ | expected rewards $\in \mathbb{R}^{SA}$ (Section 1.1); random reward $\in [-1, 1]$ (Assumption 3). |
| $\gamma \in [0, 1)$ | discount factor (Section 1.1). |
| $v, v^*, v_\pi$ | value functions $S \to \mathbb{R}$. |
| $\Delta_\mathcal{S}, \Delta_\mathcal{A}$ | sets of probability distributions over states and actions. |
| $\mu, \pi, \pi(s)$ | probability distributions in $\Delta_\mathcal{S}$ and $\Delta_\mathcal{A}$, respectively; policy in $\mathcal{S} \to \Delta_\mathcal{A}$. |
| $\Phi, \varphi_s, \varphi_0$ | feature matrix $\in \mathbb{R}^{S \times d}$; state features $\in \mathbb{R}^d$; features of planning state $\varphi_{s_0}$. |
| $\varepsilon_{\text{approx}}$ | approximation error of $\Phi$: $\min_{\theta \in \mathbb{R}^d} \|v^* - \Phi\theta\|_\infty$. |
| $W, W_*$ | constraint matrices in $\{0, 1\}^{(1+m)A \times SA}$ and $\{0, 1\}^{mA \times SA}$ (Theorems 2 and 5). |
| $\lambda, \lambda_*, \theta$ | dual variables $\in \mathbb{R}_+^{(1+m)A}$ and $\in \mathbb{R}_+^{mA}$, respectively; primal variables $\in \mathbb{R}^d$. |
| $\Lambda, \Lambda_\gamma, \mathcal{B}$ | dual spaces $\subset \mathbb{R}_+^{(1+m)A}$; primal space $\subset \mathbb{R}^d$ (Theorem 2 and Lemmas 8 and 11). |
| $\|\cdot\|_*$ | dual norm of $\|\cdot\|$: defined by $\|u\|_* = \sup_{\|x\|=1}\langle u, x\rangle$. |

## A  Proofs

### A.1  Approximation Error for the Linearly Relaxed Approximate LP

We start by recalling and improving the approximation error bounds for the Linearly Relaxed Approximate Linear Program (LRALP$_\mu$) of Lakshminarayanan et al. [27].

**Theorem 5.** *Suppose Assumptions 1 and 2 hold. Define the matrix $W_* \in \{0, 1\}^{mA \times SA}$ with rows $[W_*]_{sa} = e_{sa} \in \mathbb{R}^{SA}$ ($s \in \mathcal{S}_*, a \in \mathcal{A}$). For any (possibly unnormalized) initial distribution $\mu \in \mathbb{R}_+^S$,*

$$V_{\text{LRALP}}(\mu) := \min \{ \mu^\mathsf{T}\Phi\theta \mid \theta \in \mathbb{R}^d, \ W_* r + W_*(\gamma P - E)\Phi\theta \leq \mathbf{0} \}. \qquad \text{(LRALP}_\mu\text{)}$$

*The value of (LRALP$_\mu$) is close to the optimal value of that initial distribution:*

$$|V_{\text{LRALP}}(\mu) - \mu^\mathsf{T} v^*| \leq \frac{10\|\mu\|_1 \varepsilon_{\text{approx}}}{1 - \gamma}.$$

This result follows from Lakshminarayanan et al. [27, Theorem IV.1], which we will not reproduce here for brevity. The error bound there is $2\|\mu\|_1(3\varepsilon_{\text{approx}} + \|J_{\text{ALP}}^* - J_{\text{LRA}}^*\|_\infty)/(1 - \gamma)$, defining

$$J_{\text{ALP}}^*(s) := \min \{ \varphi_s^\mathsf{T}\theta \mid \theta \in \mathbb{R}^d, \ \Phi\theta \geq v^* \},$$

$$J_{\text{LRA}}^*(s) := \min \{ \varphi_s^\mathsf{T}\theta \mid \theta \in \mathbb{R}^d, \ W_* E\Phi\theta \geq W_* E v^* \}.$$

It only remains for us to bound $\|J_{\text{ALP}}^* - J_{\text{LRA}}^*\|_\infty$, improving upon Theorem IV.2 [27]:

**Lemma 6.** *Under the conditions of Theorem 5, $\|J_{\text{ALP}}^* - J_{\text{LRA}}^*\|_\infty \leq 2\varepsilon_{\text{approx}}$.*

*Proof.* By Assumption 1, the optimal value function is well-approximated by the feature representation; $v^* = \Phi\theta + \delta$ for some $\theta \in \mathbb{R}^d$ and $\delta \in \mathbb{R}^S$ with $\|\delta\|_\infty \leq \varepsilon_{\text{approx}}$. By Assumption 2, $\Phi = Z\Phi_*$, so $v^* = Z\Phi_*\theta + \delta$. We use these facts after writing the linear program defining $J_{\text{ALP}}^*(s)$ in its dual form:

$$J_{\text{ALP}}^*(s) = \max \{ \mu^\mathsf{T} v^* \mid \mu \in \mathbb{R}_+^S, \ \mu^\mathsf{T}\Phi = \varphi_s^\mathsf{T} \}$$

$$= \max \{ \mu^\mathsf{T}(Z\Phi_*\theta + \delta) \mid \mu \in \mathbb{R}_+^S, \ \mu^\mathsf{T} Z\Phi_* = \varphi_s^\mathsf{T} \}$$

By Assumption 1, there is some $\boldsymbol{\eta} \in \mathbb{R}^d$ such that $\boldsymbol{\Phi}\boldsymbol{\eta} = \mathbf{1}$. If $\boldsymbol{\mu}^\mathsf{T}\boldsymbol{\Phi} = \boldsymbol{\varphi}_s^\mathsf{T}$, then $\|\boldsymbol{\mu}\|_1 = \boldsymbol{\mu}^\mathsf{T}\mathbf{1} = \boldsymbol{\mu}^\mathsf{T}\boldsymbol{\Phi}\boldsymbol{\eta} = \boldsymbol{\varphi}_s^\mathsf{T}\boldsymbol{\eta} = 1$, which means that $\boldsymbol{\mu}^\mathsf{T}\boldsymbol{\delta} \leq \|\boldsymbol{\delta}\|_\infty$. Replacing $\boldsymbol{\mu}^\mathsf{T}\boldsymbol{\delta}$ with $\|\boldsymbol{\delta}\|_\infty$ in the objective increases its value; we move the resulting constant term out of the maximization:

$$\leq \|\boldsymbol{\delta}\|_\infty + \max \ \{\ \boldsymbol{\mu}^\mathsf{T} \boldsymbol{Z}\boldsymbol{\Phi}_*\boldsymbol{\theta} \mid \boldsymbol{\mu} \in \mathbb{R}_+^S, \ \boldsymbol{\mu}^\mathsf{T}\boldsymbol{Z}\boldsymbol{\Phi}_* = \boldsymbol{\varphi}_s^\mathsf{T}\ \}$$

The objective and constraints of this maximization problem depend on $\boldsymbol{\mu}$ only through $\boldsymbol{\mu}^\mathsf{T}\boldsymbol{Z}$. Thus we can replace $\boldsymbol{\mu}^\mathsf{T}\boldsymbol{Z}$ with $\boldsymbol{\mu}_* \in \mathbb{R}_+^m$, which can only expand the feasible set of the maximization and increase its value:

$$\leq \|\boldsymbol{\delta}\|_\infty + \max \ \{\ \boldsymbol{\mu}_*^\mathsf{T}\boldsymbol{\Phi}_*\boldsymbol{\theta} \mid \boldsymbol{\mu}_* \in \mathbb{R}_+^m, \ \boldsymbol{\mu}_*^\mathsf{T}\boldsymbol{\Phi}_* = \boldsymbol{\varphi}_s^\mathsf{T}\ \}$$

The matrix $\boldsymbol{U} \in \{0,1\}^{m \times S}$ with rows $[\boldsymbol{U}_s]_{s \in \mathcal{S}_*} = \boldsymbol{e}_s$ can be used to "select" the core state features from $\boldsymbol{\Phi}$, giving $\boldsymbol{\Phi}_* = \boldsymbol{U}\boldsymbol{\Phi}$:

$$= \|\boldsymbol{\delta}\|_\infty + \max \ \{\ \boldsymbol{\mu}_*^\mathsf{T}\boldsymbol{U}\boldsymbol{\Phi}\boldsymbol{\theta} \mid \boldsymbol{\mu}_* \in \mathbb{R}_+^m, \ \boldsymbol{\mu}_*^\mathsf{T}\boldsymbol{U}\boldsymbol{\Phi} = \boldsymbol{\varphi}_s^\mathsf{T}\ \}$$

By a similar argument as before, we see that $\|\boldsymbol{\mu}_*^\mathsf{T}\boldsymbol{U}\|_1 = 1$. We add $\boldsymbol{\mu}_*^\mathsf{T}\boldsymbol{U}\boldsymbol{\delta} + \|\boldsymbol{\delta}\|_\infty \geq 0$ to the objective (increasing its value), then move the constant out:

$$\leq 2\|\boldsymbol{\delta}\|_\infty + \max \ \{\ \boldsymbol{\mu}_*^\mathsf{T}\boldsymbol{U}\boldsymbol{\Phi}\boldsymbol{\theta} + \boldsymbol{\mu}_*^\mathsf{T}\boldsymbol{U}\boldsymbol{\delta} \mid \boldsymbol{\mu}_* \in \mathbb{R}_+^m, \ \boldsymbol{\mu}_*^\mathsf{T}\boldsymbol{U}\boldsymbol{\Phi} = \boldsymbol{\varphi}_s^\mathsf{T}\ \}$$
$$= 2\|\boldsymbol{\delta}\|_\infty + \max \ \{\ \boldsymbol{\mu}_*^\mathsf{T}\boldsymbol{U}\boldsymbol{v}^* \mid \boldsymbol{\mu}_* \in \mathbb{R}_+^m, \ \boldsymbol{\mu}_*^\mathsf{T}\boldsymbol{U}\boldsymbol{\Phi} = \boldsymbol{\varphi}_s^\mathsf{T}\ \}$$
$$= 2\|\boldsymbol{\delta}\|_\infty + \min \ \{\ \boldsymbol{\varphi}_s^\mathsf{T}\boldsymbol{\theta} \mid \boldsymbol{\theta} \in \mathbb{R}^d, \ \boldsymbol{U}\boldsymbol{\Phi}\boldsymbol{\theta} \geq \boldsymbol{U}\boldsymbol{v}^*\ \},$$

where the last step is obtained by writing the dual of the linear program in the previous step. Now observe that the constraint $\boldsymbol{U}\boldsymbol{\Phi}\boldsymbol{\theta} \geq \boldsymbol{U}\boldsymbol{v}^*$ is equivalent to the constraint $\boldsymbol{W}_*\boldsymbol{E}\boldsymbol{\Phi}\boldsymbol{\theta} \geq \boldsymbol{W}_*\boldsymbol{E}\boldsymbol{v}^*$ in the definition of $J_{\text{LRA}}^*$ — both of them require that $\boldsymbol{\varphi}_s\boldsymbol{\theta} \geq v_s^*$ for $s \in \mathcal{S}_*$. Thus we have shown that $J_{\text{ALP}}^*(s) - J_{\text{LRA}}^*(s) \leq 2\varepsilon_{\text{approx}}$ for all $s \in \mathcal{S}$. We also know that $J_{\text{ALP}}^*(s) \geq J_{\text{LRA}}^*(s)$, since $J_{\text{LRA}}^*(s)$ is a relaxation of $J_{\text{ALP}}^*(s)$. It follows that $\|J_{\text{ALP}}^* - J_{\text{LRA}}^*\|_\infty \leq 2\varepsilon_{\text{approx}}$. □

## A.2  Proof of Theorem 2 — Approximation Error for CoreLP

**Theorem 2** (CoreLP). *Suppose Assumptions 1 and 2 hold, $s_0 \in \mathcal{S}$, $\boldsymbol{\varphi}_0 := \boldsymbol{\varphi}_{s_0}$, $\boldsymbol{W} \in \{0,1\}^{(1+m)A \times SA}$ has rows $[\boldsymbol{W}_{s_i a}]_{s_i \in \mathcal{S}_+, a \in \mathcal{A}} = \boldsymbol{e}_{s_i a}$, and $\Lambda := \{\boldsymbol{\lambda} \in \mathbb{R}_+^{(1+m)A} \mid \sum_{a \in \mathcal{A}} \lambda_{s_0 a} = 1\}$. Define*

$$V^\dagger = \max \ \big\{\ \boldsymbol{\lambda}^\mathsf{T}\boldsymbol{W}\boldsymbol{r} \ \big| \ \boldsymbol{\lambda} \in \Lambda, \ \boldsymbol{\varphi}_0^\mathsf{T} + \boldsymbol{\lambda}^\mathsf{T}\boldsymbol{W}(\gamma\boldsymbol{P} - \boldsymbol{E})\boldsymbol{\Phi} = \mathbf{0}\ \big\}. \tag{CoreLP}$$

*Let $\boldsymbol{\lambda}^\dagger \in \Lambda$ be a maximizer of (CoreLP) and let $\boldsymbol{\pi}^\dagger \in \Delta_{\mathcal{A}}$ be given by $\pi^\dagger(a) = \lambda_{s_0 a}^\dagger$. Then*

$$|V^\dagger - v^*(s_0)| \leq \frac{10\gamma\varepsilon_{\text{approx}}}{1 - \gamma}, \qquad v^*(s_0) - \sum_{a \in \mathcal{A}} \pi^\dagger(a)\, q^*(s_0, a) \leq \frac{20\gamma\varepsilon_{\text{approx}}}{1 - \gamma}.$$

By the definition of $\Lambda \subset \mathbb{R}_+^{(1+m)A}$, we can decompose its elements as $\boldsymbol{\lambda} = \boldsymbol{\pi} \oplus \boldsymbol{\lambda}_*$, with $\boldsymbol{\pi} \in \Delta_{\mathcal{A}}$ as in the statement of the theorem and $\boldsymbol{\lambda}_* \in \mathbb{R}_+^{mA}$ defined by $\lambda_{*,sa} = \lambda_{sa}$ for $s \in \mathcal{S}_*, a \in \mathcal{A}$ — in other words, $\Lambda \cong \Delta_{\mathcal{A}} \times \mathbb{R}_+^{mA}$. The main idea of the proof is that when $\boldsymbol{\lambda}$ is a solution of (CoreLP), then $\boldsymbol{\lambda}_*$ is a solution for the dual form of (LRALP$_\mu$) from Theorem 5. To make this connection between the two problems more precise, let us write the saddle-point forms of (LRALP$_\mu$) and (CoreLP):

$$V_{\text{LRALP}}(\boldsymbol{\mu}) = \max_{\boldsymbol{\lambda}_* \in \mathbb{R}_+^{mA}} \min_{\boldsymbol{\theta} \in \mathbb{R}^d} \big[\ g_{\boldsymbol{\mu}}(\boldsymbol{\lambda}_*, \boldsymbol{\theta}) := \boldsymbol{\lambda}_*^\mathsf{T}\boldsymbol{W}_*\boldsymbol{r} + \boldsymbol{\mu}^\mathsf{T}\boldsymbol{\Phi}\boldsymbol{\theta} + \boldsymbol{\lambda}_*^\mathsf{T}\boldsymbol{W}_*(\gamma\boldsymbol{P} - \boldsymbol{E})\boldsymbol{\Phi}\boldsymbol{\theta}\ \big] \quad \text{(Saddle LRALP}_\mu\text{)}$$

$$V^\dagger = \max_{\boldsymbol{\lambda} \in \Lambda} \min_{\boldsymbol{\theta} \in \mathbb{R}^d} \big[\quad f(\boldsymbol{\lambda}, \boldsymbol{\theta}) := \boldsymbol{\lambda}^\mathsf{T}\boldsymbol{W}\boldsymbol{r} + \boldsymbol{e}_{s_0}^\mathsf{T}\boldsymbol{\Phi}\boldsymbol{\theta} + \boldsymbol{\lambda}^\mathsf{T}\boldsymbol{W}(\gamma\boldsymbol{P} - \boldsymbol{E})\boldsymbol{\Phi}\boldsymbol{\theta}\ \big] \quad \text{(Saddle CoreLP)}$$

**Lemma 7** (Corresponding (LRALP$_\mu$) and (CoreLP) solutions). *Let $\boldsymbol{\lambda} \in \Lambda \subset \mathbb{R}_+^{(1+m)A}$ be arbitrary and decompose it as $\boldsymbol{\lambda} = \boldsymbol{\pi} \oplus \boldsymbol{\lambda}_*$, where $\boldsymbol{\pi} \in \Delta_{\mathcal{A}}$ and $\boldsymbol{\lambda}_* \in \mathbb{R}_+^{mA}$. Define the distribution $\boldsymbol{\mu}_{\boldsymbol{\pi}} \in \Delta_{\mathcal{S}}$ as*

$$\boldsymbol{\mu}_{\boldsymbol{\pi}}^\mathsf{T} := \sum_{a \in \mathcal{A}} \pi(a)\, \boldsymbol{P}_{s_0 a}, \qquad\qquad \text{where } \pi(a) := \lambda_{s_0 a} \text{ for } a \in \mathcal{A}.$$

*Then, for any $\boldsymbol{\theta} \in \mathbb{R}^d$, and $g_{\boldsymbol{\mu}}(\boldsymbol{\lambda}_*, \boldsymbol{\theta})$ and $f(\boldsymbol{\lambda}, \boldsymbol{\theta})$ as in (Saddle LRALP$_\mu$) and (Saddle CoreLP),*

$$f(\boldsymbol{\lambda}, \boldsymbol{\theta}) = \sum_{a \in \mathcal{A}} \pi(a)\, r(s_0, a) + g_{\gamma\boldsymbol{\mu}_{\boldsymbol{\pi}}}(\boldsymbol{\lambda}_*, \boldsymbol{\theta}), \qquad \text{where } \lambda_{*,sa} = \lambda_{sa} \text{ for } s \in \mathcal{S}_*, a \in \mathcal{A}.$$

$P_{s_0a}$ is the next-state distribution for action $a$ at state $s_0$ — thus the distribution $\mu_\pi \in \Delta_S$ defined here is the expected next-state distribution when an action $a \sim \pi$ is taken at state $s_0$. This lemma therefore connects solutions of (CoreLP) with (LRALP$_\mu$) when $\mu = \gamma\mu_\pi$ is the discounted next-state distribution for action $a \sim \pi$.

*Proof of Lemma 7.* Recall that $W$ and $W_*$ (defined in Theorems 2 and 5) are related — the rows of $W_*$ correspond to state-action pairs in $S_* \times \mathcal{A}$, to which $W$ adds $A$ more rows corresponding to the actions at the current planning state $s_0$. Thus

$$\lambda^\mathsf{T} W = \sum_{a \in \mathcal{A}} \pi_a e_{s_0 a}^\mathsf{T} + \lambda_*^\mathsf{T} W_*, \tag{8}$$

which upon multiplying by $r$ gives

$$\lambda^\mathsf{T} W r = \sum_{a \in \mathcal{A}} \pi_a r_{s_0 a} + \lambda_*^\mathsf{T} W_* r. \tag{9}$$

Using (8) again,

$$e_{s_0}^\mathsf{T} + \lambda^\mathsf{T} W(\gamma P - E) = e_{s_0}^\mathsf{T} + \sum_{a \in \mathcal{A}} \pi_a(\gamma P_{s_0 a} - E_{s_0 a}) + \lambda_*^\mathsf{T} W_*(\gamma P - E),$$

$$= \big[ e_{s_0}^\mathsf{T} - \sum_{a \in \mathcal{A}} \pi_a E_{s_0 a} \big] + \gamma \big[ \sum_{a \in \mathcal{A}} \pi_a P_{s_0 a} \big] + \lambda_*^\mathsf{T} W_*(\gamma P - E).$$

The first term is zero because $E_{s_0 a} = e_{s_0}$ for all $a \in \mathcal{A}$, and the second term becomes $\gamma\mu_\pi$ when we substitute the definition of $\mu_\pi$. We then multiply both sides by $\Phi\theta$:

$$e_{s_0}^\mathsf{T}\Phi\theta + \lambda^\mathsf{T} W(\gamma P - E)\Phi\theta = \gamma\mu_\pi^\mathsf{T}\Phi\theta + \lambda_*^\mathsf{T} W_*(\gamma P - E)\Phi\theta.$$

Adding this to (9) gives

$$\lambda^\mathsf{T} W r + e_{s_0}^\mathsf{T}\Phi\theta + \lambda^\mathsf{T} W(\gamma P - E)\Phi\theta = \sum_{a \in \mathcal{A}} \pi_a r_{s_0 a} + \lambda_*^\mathsf{T} W_* r + \gamma\mu_\pi^\mathsf{T}\Phi\theta + \lambda_*^\mathsf{T} W_*(\gamma P - E)\Phi\theta,$$

where we substitute the definitions of $f(\lambda, \theta)$ and $g_{\gamma\mu_\pi}(\lambda_*, \theta)$ to get the desired result:

$$f(\lambda, \theta) = \sum_{a \in \mathcal{A}} \pi_a r_{s_0 a} + g_{\gamma\mu_\pi}(\lambda_*, \theta). \qquad \square$$

*Proof of Theorem 2.* Using the decomposition $\Lambda \cong \Delta_\mathcal{A} \times \mathbb{R}_+^{mA}$ in (Saddle CoreLP):

$$V^\dagger = \max_{\pi \in \Delta_\mathcal{A}} \max_{\lambda_* \in \mathbb{R}_+^{mA}} \min_{\theta \in \mathbb{R}^d} f(\pi \oplus \lambda_*, \theta) \qquad \text{(where } \pi \oplus \lambda_* = \lambda \in \Lambda)$$

$$= \max_{\pi \in \Delta_\mathcal{A}} \max_{\lambda_* \in \mathbb{R}_+^{mA}} \min_{\theta \in \mathbb{R}^d} \big[ \sum_{a \in \mathcal{A}} \pi_a r_{s_0 a} + g_{\gamma\mu_\pi}(\lambda_*, \theta) \big] \qquad \text{(using Lemma 7)}$$

$$= \max_{\pi \in \Delta_\mathcal{A}} \big[ \sum_{a \in \mathcal{A}} \pi_a r_{s_0 a} + \max_{\lambda_* \in \mathbb{R}_+^{mA}} \min_{\theta \in \mathbb{R}^d} g_{\gamma\mu_\pi}(\lambda_*, \theta) \big]$$

$$= \max_{\pi \in \Delta_\mathcal{A}} \big[ q^\dagger(\pi) \coloneqq \sum_{a \in \mathcal{A}} \pi_a r_{s_0 a} + V_{\text{LRALP}}(\gamma\mu_\pi) \big]. \qquad \text{(from (Saddle LRALP}_\mu))$$

We now turn our attention to bounding $V^\dagger$. From Theorem 5, we know that $|V_{\text{LRALP}}(\gamma\mu_\pi) - \gamma\mu_\pi^\mathsf{T} v^*| \le 10\gamma\varepsilon_{\text{approx}}/(1-\gamma)$ for any distribution over states $\mu_\pi \in \Delta_S$. Through a slight abuse of notation, we define $q^*(s_0, \pi) \coloneqq \sum_a \pi_a r_{s_0 a} + \gamma\mu_\pi^\mathsf{T} v^*$ as a generalization of the standard $q^*(s, a)$ value function to action distributions. Note that we will only need $q^*(s_0, \cdot)$, for which this abuse is 'sensible'. Then for all $\pi \in \Delta_\mathcal{A}$,

$$|q^\dagger(\pi) - q^*(s_0, \pi)| = |V_{\text{LRALP}}(\gamma\mu_\pi) - \gamma\mu_\pi^\mathsf{T} v^*| \le \frac{10\gamma\varepsilon_{\text{approx}}}{1-\gamma}. \tag{10}$$

We also know that $v^*(s_0) = \max_{\pi \in \Delta_\mathcal{A}} q^*(s_0, \pi)$ (the equality happens with $\pi^* = e_{a^*}$ for an optimal action $a^*$). Hence,

$$|V^\dagger - v^*(s_0)| = |\max_{\pi \in \Delta_\mathcal{A}} q^\dagger(\pi) - \max_{\pi \in \Delta_\mathcal{A}} q^*(s_0, \pi)| \le \max_{\pi \in \Delta_\mathcal{A}} |q^\dagger(\pi) - q^*(s_0, \pi)| \le \frac{10\gamma\varepsilon_{\text{approx}}}{1-\gamma},$$

where the last inequality follows from (10).

For the second part of the result, let $\boldsymbol{\lambda}^\dagger$ be a maximizer of (CoreLP) and $\boldsymbol{\pi}^\dagger$ be the action-distribution component (as before) so that $V^\dagger = q^\dagger(\boldsymbol{\pi}^\dagger)$. Then, using again (10), combined with the last inequality,

$$\sum_{a \in \mathcal{A}} \pi^\dagger(a)\, q^*(s_0, a) \equiv q^*(s_0, \boldsymbol{\pi}^\dagger) \geq q^\dagger(\boldsymbol{\pi}^\dagger) - \frac{10\gamma\varepsilon_{\text{approx}}}{1-\gamma}$$

$$= V^\dagger - \frac{10\gamma\varepsilon_{\text{approx}}}{1-\gamma}$$

$$\geq v^*(s_0) - \frac{20\gamma\varepsilon_{\text{approx}}}{1-\gamma}.$$

Reordering gives the desired result, namely that $v^*(s_0) - \sum_a \pi^\dagger(a)\, q^*(s_0, a) \leq 20\gamma\varepsilon_{\text{approx}}/(1-\gamma)$. □

## A.3 Proof of Theorem 3 — Error Bounds for the CoreStoMP Algorithm

**Theorem 3** (CoreStoMP). *Suppose Assumptions 1, 2, and 3 hold, and define*

$$B := \frac{(9/8)\sqrt{m}}{1-\gamma}, \qquad\qquad C := \frac{(9/4)\sqrt{m(1 + 2\log A + 2\gamma\log m)}}{(1-\gamma)^2}.$$

*Let $\hat{\boldsymbol{\lambda}}$ be the result of running Algorithm 1 for T iterations with the parameter B and the step size $\eta = C^{-1}\sqrt{2/7T}$, which requires $2T(1 + (1+m)A)$ simulator queries. Define $\hat{\boldsymbol{\pi}} \in \Delta_{\mathcal{A}}$ by $\hat{\pi}(a) = \hat{\lambda}_{s_0 a}$ (as in Theorem 2) and $a \sim \hat{\boldsymbol{\pi}}$. Then*

$$v^*(s_0) - \mathbb{E}[q^*(s_0, a)] \leq \frac{32\varepsilon_{\text{approx}}}{1-\gamma} + \frac{21}{2(1-\gamma)^2}\sqrt{\frac{3m(1 + 2\log A + 2\gamma\log m)}{T}}.$$

The proof of this theorem has two main ingredients: First, in Lemma 8, we show that approximate solutions of (Saddle CoreLP) can be used to recover near-optimal action distributions for the planning state $s_0$ — the approximation quality is measured by the *duality gap*. Second, in Lemma 11, we bound the expected duality gap of the Stochastic Mirror-Prox algorithm when specialized to our setting.

**Lemma 8** (Approximate (Saddle CoreLP) solutions). *Suppose $\mathcal{B} \subset \mathbb{R}^d$ and $C_{\mathcal{B}} \geq 0$ are chosen such that, for any distribution over states $\boldsymbol{\mu} \in \Delta_{\mathcal{S}}$, there is some $\boldsymbol{\theta} \in \mathcal{B}$ that is feasible for (LRALP$_\mu$) and at most $C_{\mathcal{B}}$-suboptimal. Define*

$$\Lambda_\gamma := \{\boldsymbol{\lambda} \in \Lambda \mid \|\boldsymbol{\lambda}\|_1 = 1/(1-\gamma)\}, \tag{11}$$

*a subset of the set $\Lambda \subset \mathbb{R}_+^{(1+m)A}$ from Theorem 2. Define the $\mathcal{B}$-bounded duality gap of an approximate solution of (Saddle CoreLP) as*

$$\delta_{\mathcal{B}}(\hat{\boldsymbol{\lambda}}, \hat{\boldsymbol{\theta}}) := \max_{\boldsymbol{\lambda} \in \Lambda_\gamma} f(\boldsymbol{\lambda}, \hat{\boldsymbol{\theta}}) - \inf_{\boldsymbol{\theta} \in \mathcal{B}} f(\hat{\boldsymbol{\lambda}}, \boldsymbol{\theta}), \qquad\qquad \text{where } \hat{\boldsymbol{\lambda}} \in \Lambda \text{ and } \hat{\boldsymbol{\theta}} \in \mathbb{R}^d. \tag{12}$$

*For any $\hat{\boldsymbol{\lambda}} \in \Lambda$ and $\hat{\boldsymbol{\theta}} \in \mathbb{R}^d$, let $\hat{\boldsymbol{\pi}}$ be the action distribution component of $\hat{\boldsymbol{\lambda}}$, as in Theorem 2. Then*

$$v^*(s_0) - \sum_{a \in \mathcal{A}} \hat{\pi}(a)\, q^*(s_0, a) \leq \frac{20\gamma\varepsilon_{\text{approx}}}{1-\gamma} + \gamma C_{\mathcal{B}} + \delta_{\mathcal{B}}(\hat{\boldsymbol{\lambda}}, \hat{\boldsymbol{\theta}}).$$

This lemma generalizes the second result of Theorem 2 in two ways: First, the Stochastic Mirror-Prox algorithm does not produce exact solutions of (Saddle CoreLP); the optimization error is measured by the duality gap — here we see the effect of a non-zero duality gap on the resulting action distribution. Second, the primal variables $\boldsymbol{\theta}$ in (Saddle CoreLP) have the unbounded domain $\mathbb{R}^d$, whereas the Stochastic Mirror-Prox algorithm requires the optimization domain to have a bounded *diameter*; see Proof of Lemma 11 — Stochastic Mirror-Prox. This lemma shows that restricting $\boldsymbol{\theta}$ to a large-enough bounded set $\mathcal{B}$ only incurs an additional $C_{\mathcal{B}}$ error. Indeed, the second issue is related to the first — an *unbounded* form of the duality gap would be infinite for *any* approximate solution, making it useless as a measure of optimization accuracy; the $\mathcal{B}$-bounded duality gap therefore addresses both these issues:

*Claim* 9. For any $\hat{\boldsymbol{\lambda}} \in \Lambda$, $\hat{\boldsymbol{\theta}} \in \mathbb{R}^d$, $\boldsymbol{\theta} \in \mathcal{B} \subset \mathbb{R}^d$, and $\delta_{\mathcal{B}}(\hat{\boldsymbol{\lambda}}, \hat{\boldsymbol{\theta}})$ being the $\mathcal{B}$-bounded duality gap (12),

$$V^\dagger \leq f(\hat{\boldsymbol{\lambda}}, \boldsymbol{\theta}) + \delta_{\mathcal{B}}(\hat{\boldsymbol{\lambda}}, \hat{\boldsymbol{\theta}}).$$

*Proof.* Let $\boldsymbol{\lambda}^* \in \Lambda \subset \mathbb{R}_+^{(1+m)A}$ be a maximizer of (CoreLP) — this exists because the optimization is bounded (Theorem 2). Then

$$\mathbf{0} = \boldsymbol{\varphi}_0^\mathsf{T} + \boldsymbol{\lambda}^{*\mathsf{T}} \boldsymbol{W}(\gamma \boldsymbol{P} - \boldsymbol{E})\boldsymbol{\Phi} \qquad \text{(since } \boldsymbol{\lambda}^* \text{ is feasible for (CoreLP))} \qquad (13)$$

$$= \boldsymbol{\varphi}_0^\mathsf{T}\hat{\boldsymbol{\theta}} + \boldsymbol{\lambda}^{*\mathsf{T}} \boldsymbol{W}(\gamma \boldsymbol{P} - \boldsymbol{E})\boldsymbol{\Phi}\hat{\boldsymbol{\theta}}. \qquad \text{(multiplying by } \hat{\boldsymbol{\theta}} \in \mathbb{R}^d)$$

Since, $\boldsymbol{\lambda}^*$ is a maximizer of (CoreLP), $V^\dagger = \boldsymbol{\lambda}^{*\mathsf{T}}\boldsymbol{W}\boldsymbol{r}$:

$$V^\dagger = \boldsymbol{\lambda}^{*\mathsf{T}}\boldsymbol{W}\boldsymbol{r} + \boldsymbol{\varphi}_0^\mathsf{T}\hat{\boldsymbol{\theta}} + \boldsymbol{\lambda}^{*\mathsf{T}} \boldsymbol{W}(\gamma \boldsymbol{P} - \boldsymbol{E})\boldsymbol{\Phi}\hat{\boldsymbol{\theta}} \qquad \text{(adding } V^\dagger \text{ on l.h.s. and } \boldsymbol{\lambda}^{*\mathsf{T}}\boldsymbol{W}\boldsymbol{r} \text{ on r.h.s.)}$$

$$= f(\boldsymbol{\lambda}^*, \hat{\boldsymbol{\theta}}). \qquad \text{(definition of } f \text{ from (Saddle CoreLP))} \qquad (14)$$

Assumption 1 tells us that $\boldsymbol{\Phi}\boldsymbol{\eta} = \mathbf{1}$ for some $\boldsymbol{\eta} \in \mathbb{R}^d$ — multiplying (13) by $\boldsymbol{\eta}$, we see that $\boldsymbol{\lambda}^*$ must satisfy $1 + \gamma\|\boldsymbol{\lambda}^*\|_1 = \|\boldsymbol{\lambda}^*\|_1$, as does any other feasible solution of (CoreLP). In particular, this means that $\|\boldsymbol{\lambda}^*\|_1 = 1/(1-\gamma)$ and so $\boldsymbol{\lambda}^* \in \Lambda_\gamma$. Using the definition of $\delta_{\mathcal{B}}$ from (12),

$$\delta(\hat{\boldsymbol{\lambda}}, \hat{\boldsymbol{\theta}}) \geq f(\boldsymbol{\lambda}^*, \hat{\boldsymbol{\theta}}) - f(\hat{\boldsymbol{\lambda}}, \boldsymbol{\theta}) \qquad \text{(since } \boldsymbol{\lambda}^* \in \Lambda_\gamma \text{ and } \boldsymbol{\theta} \in \mathcal{B})$$

$$= V^\dagger - f(\hat{\boldsymbol{\lambda}}, \boldsymbol{\theta}). \qquad \text{(using (14))} \qquad \square$$

*Claim* 10. For any $\hat{\boldsymbol{\lambda}}_* \in \mathbb{R}_+^{mA}$ and distribution over states $\boldsymbol{\mu} \in \Delta_{\mathcal{S}}$, suppose $\boldsymbol{\theta} \in \mathbb{R}^d$ is feasible for (LRALP$_\mu$) and at most $C_{\mathcal{B}}$-suboptimal. Then, with $g_\mu$ being the objective function of (Saddle LRALP$_\mu$),

$$V_{\text{LRALP}}(\gamma\boldsymbol{\mu}) \geq g_{\gamma\boldsymbol{\mu}}(\hat{\boldsymbol{\lambda}}_*, \boldsymbol{\theta}) - \gamma C_{\mathcal{B}}.$$

*Proof.* Since $\boldsymbol{\theta}$ is feasible for (LRALP$_\mu$), $\boldsymbol{W}_*\boldsymbol{r} + \boldsymbol{W}_*(\gamma\boldsymbol{P} - \boldsymbol{E})\boldsymbol{\Phi}\boldsymbol{\theta} \leq \mathbf{0}$. Multiplying both sides of this inequality by $\hat{\boldsymbol{\lambda}}_*^\mathsf{T} \geq \mathbf{0}$,

$$\hat{\boldsymbol{\lambda}}_*^\mathsf{T}\boldsymbol{W}_*\boldsymbol{r} + \hat{\boldsymbol{\lambda}}_*^\mathsf{T}\boldsymbol{W}_*(\gamma\boldsymbol{P} - \boldsymbol{E})\boldsymbol{\Phi}\boldsymbol{\theta} \leq 0$$

$$\hat{\boldsymbol{\lambda}}_*^\mathsf{T}\boldsymbol{W}_*\boldsymbol{r} + \gamma\boldsymbol{\mu}^\mathsf{T}\boldsymbol{\Phi}\boldsymbol{\theta} + \hat{\boldsymbol{\lambda}}_*^\mathsf{T}\boldsymbol{W}_*(\gamma\boldsymbol{P} - \boldsymbol{E})\boldsymbol{\Phi}\boldsymbol{\theta} \leq \gamma\boldsymbol{\mu}^\mathsf{T}\boldsymbol{\Phi}\boldsymbol{\theta} \qquad \text{(adding } \gamma\boldsymbol{\mu}^\mathsf{T}\boldsymbol{\Phi}\boldsymbol{\theta} \text{ to both sides)}$$

$$g_{\gamma\boldsymbol{\mu}}(\hat{\boldsymbol{\lambda}}_*, \boldsymbol{\theta}) \leq \gamma\boldsymbol{\mu}^\mathsf{T}\boldsymbol{\Phi}\boldsymbol{\theta}. \qquad \text{(definition of } g \text{ from (Saddle LRALP}_\mu))$$

Note that the choice of $\boldsymbol{\mu}$ does not affect the constraints of (LRALP$_\mu$), only its objective function — thus $\boldsymbol{\theta}$ is feasible for the problem defining $V_{\text{LRALP}}(\gamma\boldsymbol{\mu})$ and is $\gamma C_{\mathcal{B}}$ suboptimal: $\gamma\boldsymbol{\mu}^\mathsf{T}\boldsymbol{\Phi}\boldsymbol{\theta} \leq V_{\text{LRALP}}(\gamma\boldsymbol{\mu}) + \gamma C_{\mathcal{B}}$. Substituting this into the last inequality and rearranging gives the desired result. $\square$

*Proof of Lemma 8.* As in Lemma 7, we write $\hat{\boldsymbol{\lambda}} = \hat{\boldsymbol{\pi}} \oplus \hat{\boldsymbol{\lambda}}_*$ with $\hat{\boldsymbol{\pi}} \in \Delta_{\mathcal{A}}$ and $\hat{\boldsymbol{\lambda}}_* \in \mathbb{R}_+^{mA}$ and define $\boldsymbol{\mu}_{\hat{\boldsymbol{\pi}}} = \sum_a \hat{\pi}_a \boldsymbol{P}_{s_0 a}$. By our assumption, there is some $\boldsymbol{\theta} \in \mathcal{B}$ that is feasible and at most $C_{\mathcal{B}}$-suboptimal for (LRALP$_\mu$) with $\boldsymbol{\mu} = \boldsymbol{\mu}_{\hat{\boldsymbol{\pi}}}$ — this allows us to apply Claims 9 and 10 below:

$$v^*(s_0) \leq V^\dagger + \frac{10\gamma\varepsilon_{\text{approx}}}{1-\gamma} \qquad \text{(Theorem 2)}$$

$$\leq f(\hat{\boldsymbol{\lambda}}, \boldsymbol{\theta}) + \delta_{\mathcal{B}}(\hat{\boldsymbol{\lambda}}, \hat{\boldsymbol{\theta}}) + \frac{10\gamma\varepsilon_{\text{approx}}}{1-\gamma} \qquad \text{(Claim 9)}$$

$$= \sum_{a \in \mathcal{A}} \hat{\pi}_a r_{s_0 a} + g_{\gamma\boldsymbol{\mu}_{\hat{\boldsymbol{\pi}}}}(\hat{\boldsymbol{\lambda}}_*, \boldsymbol{\theta}) + \delta_{\mathcal{B}}(\hat{\boldsymbol{\lambda}}, \hat{\boldsymbol{\theta}}) + \frac{10\gamma\varepsilon_{\text{approx}}}{1-\gamma} \qquad \text{(Lemma 7)}$$

$$\leq \sum_{a \in \mathcal{A}} \hat{\pi}_a r_{s_0 a} + V_{\text{LRALP}}(\gamma\boldsymbol{\mu}_{\hat{\boldsymbol{\pi}}}) + \gamma C_{\mathcal{B}} + \delta_{\mathcal{B}}(\hat{\boldsymbol{\lambda}}, \hat{\boldsymbol{\theta}}) + \frac{10\gamma\varepsilon_{\text{approx}}}{1-\gamma} \qquad \text{(Claim 10)}$$

$$\leq \sum_{a \in \mathcal{A}} \hat{\pi}_a r_{s_0 a} + \gamma\boldsymbol{\mu}_{\hat{\boldsymbol{\pi}}}^\mathsf{T}\boldsymbol{v}^* + \gamma C_{\mathcal{B}} + \delta_{\mathcal{B}}(\hat{\boldsymbol{\lambda}}, \hat{\boldsymbol{\theta}}) + \frac{20\gamma\varepsilon_{\text{approx}}}{1-\gamma} \qquad \text{(Theorem 5)}$$

$$= \sum_{a \in \mathcal{A}} \hat{\pi}_a q^*(s_0, a) + \gamma C_{\mathcal{B}} + \delta_{\mathcal{B}}(\hat{\boldsymbol{\lambda}}, \hat{\boldsymbol{\theta}}) + \frac{20\gamma\varepsilon_{\text{approx}}}{1-\gamma},$$

where the last step used $\sum_a \hat{\pi}_a r_{s_0 a} + \gamma\boldsymbol{\mu}_{\hat{\boldsymbol{\pi}}}^\mathsf{T}\boldsymbol{v}^* = \sum_a \hat{\pi}_a(r_{s_0 a} + \gamma\boldsymbol{P}_{s_0 a}\boldsymbol{v}^*) = \sum_a \hat{\pi}_a q^*(s_0, a)$. Rearranging the inequality completes the proof:

$$v^*(s_0) - \sum_{a \in \mathcal{A}} \hat{\pi}_a q^*(s_0, a) \leq \frac{20\gamma\varepsilon_{\text{approx}}}{1-\gamma} + \gamma C_{\mathcal{B}} + \delta_{\mathcal{B}}(\hat{\boldsymbol{\lambda}}, \hat{\boldsymbol{\theta}}). \qquad \square$$

The following lemma bounds the expected duality gap of the Stochastic Mirror-Prox algorithm when applied to our setting. We defer the proof to Appendix A.4.

**Lemma 11** (Stochastic Mirror-Prox). *Using the constants B and C from Theorem 3, define*

$$\mathcal{B} := \{\boldsymbol{\theta} \in \mathbb{R}^d \mid \|\boldsymbol{\Phi}_*\boldsymbol{\theta}\|_2 \leq B\}, \tag{15}$$

*and let $\delta_{\mathcal{B}}(\hat{\boldsymbol{\lambda}}, \hat{\boldsymbol{\theta}})$ be the $\mathcal{B}$-bounded duality gap (12). Then the results of running Algorithm 1 for T iterations satisfy*

$$\varepsilon_{\text{opt}} := \mathbb{E}[\delta(\hat{\boldsymbol{\lambda}}, \hat{\boldsymbol{\theta}})] \leq \frac{14C}{\sqrt{3T}}.$$

*Proof of Theorem 3.* First, observe that $v^*(s_0) - q^*(s_0, a) \leq 2/(1 - \gamma)$ for any action $a$, since all the rewards lie in $[-1, 1]$ by Assumption 3. Thus, if $\varepsilon_{\text{approx}} > 1/16$ then $32\varepsilon_{\text{approx}}/(1 - \gamma) > 2/(1 - \gamma)$ and the result is trivially true. From now on, we will assume that $\varepsilon_{\text{approx}} \leq 1/16$.

To prove our result, we will combine Lemmas 8 and 11, for which we need to show that the set $\mathcal{B}$ defined in (15) satisfies the requirements of Lemma 8. Specifically, we need to show that $\mathcal{B}$ contains a feasible solution of the linear program (LRALP$_\mu$) with sub-optimality bounded by a constant $C_{\mathcal{B}}$. Note that the constraints of (LRALP$_\mu$) do not depend on $\mu$, only the objective function, so the choice of $\mu$ does not affect feasibility.

Since (LRALP$_\mu$) is a relaxation of the ALP (see Section 1.2), any feasible solution of the ALP is also feasible for (LRALP$_\mu$). De Farias and Van Roy [14, Theorem 2] show that the ALP has a feasible solution $\boldsymbol{\Phi}\boldsymbol{\theta}$ that is close to $v^*$ — more precisely

$$\|\boldsymbol{\Phi}\boldsymbol{\theta} - v^*\|_\infty \leq \frac{2\varepsilon_{\text{approx}}}{1 - \gamma}. \tag{16}$$

Since $\|v^*\|_\infty \leq 1/(1 - \gamma)$, we must have $\|\boldsymbol{\Phi}\boldsymbol{\theta}\|_\infty \leq (1 + 2\varepsilon_{\text{approx}})/(1 - \gamma)$. It follows that

$$\|\boldsymbol{\Phi}_*\boldsymbol{\theta}\|_2 \leq \sqrt{m}\|\boldsymbol{\Phi}_*\boldsymbol{\theta}\|_\infty = \sqrt{m}\|\boldsymbol{\Phi}\boldsymbol{\theta}\|_\infty \leq \frac{(1 + 2\varepsilon_{\text{approx}})\sqrt{m}}{1 - \gamma} \leq \frac{(9/8)\sqrt{m}}{1 - \gamma} = B,$$

where the first inequality is a property of the 2-norm, the next equality is thanks to Assumption 2, and the last inequality is because we assumed $\varepsilon_{\text{approx}} \leq 1/16$ — this shows that $\boldsymbol{\theta} \in \mathcal{B}$. We also have

$$\boldsymbol{\mu}^{\mathsf{T}}\boldsymbol{\Phi}\boldsymbol{\theta} - \boldsymbol{\mu}^{\mathsf{T}}v^* \leq \|\boldsymbol{\mu}\|_1\|\boldsymbol{\Phi}\boldsymbol{\theta} - v^*\|_\infty \leq \frac{2\varepsilon_{\text{approx}}}{1 - \gamma} \qquad \text{(using (16) and } \|\boldsymbol{\mu}\|_1 = 1)$$

$$|\boldsymbol{\mu}^{\mathsf{T}}v^* - V_{\text{LRALP}}(\boldsymbol{\mu})| \leq \frac{10\varepsilon_{\text{approx}}}{1 - \gamma} \qquad \text{(using Theorem 5 and } \|\boldsymbol{\mu}\|_1 = 1)$$

We get the value of $C_{\mathcal{B}}$ by putting these two bounds together:

$$\boldsymbol{\mu}^{\mathsf{T}}\boldsymbol{\Phi}\boldsymbol{\theta} - V_{\text{LRALP}}(\boldsymbol{\mu}) \leq \frac{12\varepsilon_{\text{approx}}}{1 - \gamma} =: C_{\mathcal{B}}.$$

Lemma 8 applied to $\mathcal{B}$ with this value of $C_{\mathcal{B}}$ gives

$$v^*(s_0) - \sum_{a \in \mathcal{A}} \hat{\pi}(a)\, q^*(s_0, a) \leq \frac{32\gamma\varepsilon_{\text{approx}}}{1 - \gamma} + \delta_{\mathcal{B}}(\hat{\boldsymbol{\lambda}}, \hat{\boldsymbol{\theta}}).$$

Taking expectations on both sides and substituting the value of $\varepsilon_{\text{opt}} = \mathbb{E}[\delta_{\mathcal{B}}(\hat{\boldsymbol{\lambda}}, \hat{\boldsymbol{\theta}})]$ from Lemma 11,

$$v^*(s_0) - \mathbb{E}[q^*(s_0, a)] \leq \frac{32\gamma\varepsilon_{\text{approx}}}{1 - \gamma} + \frac{14C}{\sqrt{3T}}.$$

We drop the $\gamma$ factor from the leading term and plug in the value of $C$ from the statement of Theorem 3 to finish the proof. □

## A.4  Proof of Lemma 11 — Stochastic Mirror-Prox

**Lemma 11** (Stochastic Mirror-Prox). *Using the constants B and C from Theorem 3, define*

$$\mathcal{B} := \{\boldsymbol{\theta} \in \mathbb{R}^d \mid \|\boldsymbol{\Phi}_*\boldsymbol{\theta}\|_2 \leq B\}, \tag{15}$$

*and let $\delta_{\mathcal{B}}(\hat{\boldsymbol{\lambda}}, \hat{\boldsymbol{\theta}})$ be the $\mathcal{B}$-bounded duality gap (12). Then the results of running Algorithm 1 for T iterations satisfy*

$$\varepsilon_{\text{opt}} := \mathbb{E}[\delta(\hat{\boldsymbol{\lambda}}, \hat{\boldsymbol{\theta}})] \leq \frac{14C}{\sqrt{3T}}.$$

Throughout this section, we will use the definitions of $\Lambda_\gamma$ and $\mathcal{B}$ from (11) and (15), respectively. We will also define the composite space $Z := \Lambda_\gamma \times \mathcal{B}$. We will use the norm $\|\lambda\|_1$ for $\lambda \in \Lambda_\gamma$, whose dual norm is $\|\cdot\|_\infty$. For $\theta \in \mathcal{B}$ we will use the norm $\|\theta\| \equiv \|\Phi\theta\|_2$ — the corresponding dual norm enjoys the convenient bound $\|\Phi_*^\mathsf{T} u\|_* = \sup_{\|\theta\|\leq 1} u^\mathsf{T}\Phi_*\theta \leq \|u\|_2 \leq \|u\|_1$ for any vector $u$.[4] The last inequality is due a general property of $p$-norms: $\|u\|_p \leq \|u\|_q$ whenever $\infty \geq p \geq q \geq 1$.

### A.4.1 Lipschitz Constants

Our first step will be to bound the Lipschitz constants associated with $f$, the objective function of (CoreLP). In other words, we are looking for bounds on $\|f_\lambda(\theta)\|_\infty$ and $\|f_\theta(\lambda)\|_*$.

First, for any $\theta \in \mathcal{B}$ we have

$$
\begin{aligned}
\|f_\lambda(\theta)\|_\infty &= \|Wr + W(\gamma P - E)\Phi\theta\|_\infty \\
&\leq \|Wr\|_\infty + \|W(\gamma P - E)\Phi\theta\|_\infty && \text{(by the triangle inequality)} \\
&\leq 1 + \max_i \|W_i(\gamma P - E)\|_1 \|\Phi\theta\|_\infty. && \text{(by definition of } \|\cdot\|_\infty \text{ and Hölder's inequality)}
\end{aligned}
$$

Now, by the property of norms, $\|\Phi\theta\|_\infty \leq \|\Phi\theta\|_2 \leq B$. Secondly, $W_i P$ and $W_i E$ are probability distributions, so $\|W_i(\gamma P - E)\|_1 \leq \gamma\|W_i P\|_1 + \|W_i E\|_1 = 1 + \gamma$ and

$$
\|f_\lambda(\theta)\|_\infty \leq 1 + (1 + \gamma)B \leq 2B. \qquad\qquad \text{(since } B \geq 1/(1-\gamma))
$$

For the other gradient, we use the bound on dual norms mentioned above:

$$
\begin{aligned}
\|f_\theta(\lambda)\|_* &= \|(\varphi_{s_0}^\mathsf{T} + \lambda^\mathsf{T} W(\gamma P - E))\Phi\|_* \\
&\leq \|e_{s_0}^\mathsf{T}\|_1 + \|\lambda^\mathsf{T} W(\gamma P - E)\|_1 \\
&\leq 1 + \frac{1+\gamma}{1-\gamma} = \frac{2}{1-\gamma},
\end{aligned}
$$

where the last inequality uses the fact that $(1-\gamma)\lambda$ is a probability distribution, as are the rows of $W$, $P$, and $E$.

### A.4.2 Gradient Estimator Variance

Next, we will bound the variance in the stochastic estimators $\hat{f}_\lambda(\theta)$ and $\hat{f}_\theta(\lambda)$ defined in (6) and (7), respectively, compared to the true gradients $f_\lambda(\theta)$ and $f_\theta(\lambda)$ defined in (4) and (5), respectively.

First, we bound $\mathbb{E}[\|\hat{f}_\lambda(\theta) - f_\lambda(\theta)\|_\infty^2]$ for any $\theta \in \mathcal{B}$ by bounding its components. For any state $s \in \mathcal{S}_+$, action $a \in \mathcal{A}$, and reward $\hat{r}$, we have

$$
\begin{aligned}
&|[\hat{f}_\lambda(\theta)]_{sa} - [f_\lambda(\theta)]_{sa}| \\
&= |(\hat{r} + \gamma\varphi_{s'}^\mathsf{T}\theta - \varphi_s^\mathsf{T}\theta) - (r_{sa} + \gamma P_{sa}\Phi\theta - \varphi_s^\mathsf{T}\theta)| && \text{(for some random } s' \sim P_{sa}) \\
&\leq |\hat{r} - r_{sa}| + \gamma|\varphi_{s'}^\mathsf{T}\theta - P_{sa}\Phi\theta| && \text{(by the triangle inequality)} \\
&\leq 2 + \gamma|(e_{s'}^\mathsf{T} - P_{sa})\Phi\theta| && \text{(bounded rewards)} \\
&\leq 2 + \gamma\|e_{s'}^\mathsf{T} - P_{sa}\|_1 \|\Phi\theta\|_\infty && \text{(using Hölder's inequality)} \\
&\leq 2 + 2\gamma B && \text{(since } \|\Phi\theta\|_\infty \leq \|\Phi\theta\|_2 \leq B) \\
&\leq 2B. && \text{(since } B \geq 1/(1-\gamma))
\end{aligned}
$$

It follows that $\|\hat{f}_\lambda(\theta) - f_\lambda(\theta)\|_\infty^2 \leq (2B)^2$, and the same bound must hold for the expectation.

We will now bound the other gradient, using the following property of Euclidean norms: for any vector-valued random variable $h$ with mean $\bar{h}$, $\mathbb{E}[\|h - \bar{h}\|^2] = \mathbb{E}[\|h\|^2] - \|\bar{h}\|^2 \leq \mathbb{E}[\|h\|^2]$. Then, for a random choice of state $s \in \mathcal{S}_+$, action $a \in \mathcal{A}$, and next state $s' \sim P_{sa}$:

$$
\begin{aligned}
\mathbb{E}[\|\hat{f}_\theta(\lambda) - f_\theta(\lambda)\|_*^2] &= \mathbb{E}\left[\left\|(\varphi_0^\mathsf{T} + \|\lambda\|_1(\gamma\varphi_{s'} - \varphi_s)) - (\varphi_0^\mathsf{T} + \lambda^\mathsf{T} W(\gamma P - E)\Phi)\right\|_*^2\right] \\
&= \left(\frac{1}{1-\gamma}\right)^2 \mathbb{E}\left[\left\|(\gamma\varphi_{s'} - \varphi_s) - (\lambda/\|\lambda\|_1)^\mathsf{T} W(\gamma P - E)\Phi\right\|_*^2\right]
\end{aligned}
$$

Now, since $(s, a) \sim \lambda/\|\lambda\|_1$ and $s' \sim P_{sa}$, we have $\mathbb{E}[\gamma\boldsymbol{\varphi}_{s'} - \boldsymbol{\varphi}_s] = (\lambda/\|\lambda\|_1)^{\mathsf{T}} W(\gamma P - E)\Phi$, so by the above property of variance for vector-valued random variables,

$$\leq \left(\frac{1}{1-\gamma}\right)^2 \mathbb{E}[\|\gamma\boldsymbol{\varphi}_{s'} - \boldsymbol{\varphi}_s\|_*^2]$$

$$= \left(\frac{1}{1-\gamma}\right)^2 \mathbb{E}[\|(\gamma\boldsymbol{e}_{s'} - \boldsymbol{e}_s)\Phi\|_*^2]$$

$$\leq \left(\frac{1+\gamma}{1-\gamma}\right)^2 \leq \left(\frac{2}{1-\gamma}\right)^2.$$

### A.4.3 Distance-Generating Functions

The Stochastic Mirror-Prox algorithm requires strongly convex *distance-generating functions* for $\Lambda_\gamma$ and $\mathcal{B}$ with respect to their respective norms. A function $\omega : \mathcal{X} \to \mathbb{R}$ (with domain $\mathcal{X} \subset \mathbb{R}^n$) is said to be $\sigma$-*strongly convex* (where $\sigma > 0$ is called the modulus of convexity) with respect to a norm $\|\cdot\|$ on $\mathcal{X}$ if any of the following conditions hold for all $x, y \in \mathcal{X}$

   (i) For all $\alpha \in [0, 1]$, $\alpha\omega(x) + (1 - \alpha)\omega(y) \geq \omega(\alpha x + (1 - \alpha)y) + \sigma\alpha(1 - \alpha)\|x - y\|^2/2$.

  (ii) $\omega$ is convex and $\omega(x) \geq \omega(y) + \langle\nabla\omega(y), x - y\rangle + \sigma\|x - y\|^2/2$.

 (iii) $\mathcal{X}$ is convex and $\langle\nabla\omega(x) - \nabla\omega(y), x - y\rangle \geq \sigma\|x - y\|^2$.

Condition (i) is the definition of strong convexity; note that it reduces to convexity when $\sigma = 0$. Conditions (ii) and (iii) are equivalent to the definition under appropriate differentiability conditions on $\omega$ that hold in our setting and when $x, y$ are in the interior of $\mathcal{X}$; see Yu [44] for details. Juditsky et al. [22] uses "strongly convex" to mean that a function is 1-strongly convex according to condition (iii).

Define the *divergence function:*

$$D_\omega(x, y) := \omega(x) - \omega(y) - \langle\nabla\omega(y), x - y\rangle.$$

When $\omega$ is convex, one can see that $D_\omega$ is always non-negative, and by condition (ii) the $\sigma$-strong convexity of $\omega$ is equivalent to $D_\omega(x, y) \geq \sigma\|x - y\|^2/2$. We will use this equivalence to establish the strong convexity of our distance-generating functions below. An important operation related to distance-generating functions is the *proximal projection* onto $\mathcal{X}$ with respect to $\omega$:

$$\Pi_\omega(x, \xi) := \arg\min_{y \in \mathcal{X}} D_\omega(y, x) + \langle\xi, y\rangle, \quad \xi \in \mathbb{R}^n.$$

The *center* of $\mathcal{X}$ with respect to $\omega$ is defined as $x_0 := \arg\min_{x \in \mathcal{X}} \omega(x)$ and the *diameter* of $\mathcal{X}$ is $\Omega_\mathcal{X} := \sup_{y \in \mathcal{X}} \sqrt{2D_\omega(y, x_0)}$.

A strongly convex distance-generating function can be thought of as a generalization of the squared norm $\|\cdot\|^2$ — the corresponding divergence generalizes the squared distance function $\|x - y\|^2$; unlike the squared distance, however, the divergence may not be symmetric. Indeed, when $\|\cdot\|$ is an Euclidean norm, and *only* for such norms, the function $\omega(x) = \|x\|^2/2$ is 1-strongly convex [44, Proposition 2]. In this special case, the divergence is $D_\omega(x, y) = \|x - y\|^2/2$ and the proximal projection is simply the Euclidean projection: $\Pi_\omega(x, \xi) = \arg\min_{y \in \mathcal{X}} \|x + \xi - y\|$.

Thus, since the domain $\mathcal{B}$ of the primal variables $\theta$ is equipped with the Euclidean norm $\|\theta\| \equiv \|\Phi\theta\|_2$, we will use the 1-strongly convex distance-generating function $\omega_\mathcal{B}(\theta) = \|\theta\|^2/2$. Since $\mathcal{B}$ is the Euclidean ball under this norm, the center of $\mathcal{B}$ is $\theta_0 = \mathbf{0}$ and its "diameter" (actually the radius, in this case) is $\Omega_\mathcal{B} = B$; $\theta_0$ is used as the initial value of $\theta$ in Algorithm 1. The proximal projection is

$$\Pi_\mathcal{B}(\theta, \xi) = \arg\min_{\theta' \in \mathcal{B}} \|\theta + \xi - \theta'\| = \frac{\theta + \xi}{\max\{1, \|\theta + \xi\|/B\}}.$$

For the dual variables $\lambda \in \Lambda_\gamma$, our distance-generating function is a modification of the *unnormalized negentropy* $h(\lambda) = \sum_i \lambda_i(\log\lambda_i - 1)$. It is well-known that this function is 1-strongly convex on the set $\{\lambda \geq \mathbf{0} \mid \|\lambda\|_1 \leq 1\}$ [e.g., 44, Theorem 5]. To achieve 1-strong convexity on $\Lambda_\gamma$ (where $\|\lambda\|_1 = 1/(1 - \gamma) > 1$), we use a modified form of this function:

$$h_\gamma(\lambda) := \frac{h((1 - \gamma)\lambda)}{(1 - \gamma)^2}.$$

It follows that $\nabla h_\gamma(\lambda) = [\nabla h((1-\gamma)\lambda)]/(1-\gamma)$. Thus, defining $D_\Lambda(\lambda', \lambda) := D_{h_\gamma}(\lambda', \lambda)$, we have

$$D_\Lambda(\lambda', \lambda) = \left(\frac{1}{1-\gamma}\right)\left[\frac{h((1-\gamma)\lambda')}{1-\gamma} - \frac{h((1-\gamma)\lambda)}{1-\gamma} - \langle \nabla h((1-\gamma)\lambda), \lambda' - \lambda \rangle\right]$$

$$= \left(\frac{1}{1-\gamma}\right)\left[\frac{h((1-\gamma)\lambda')}{1-\gamma} - \frac{h((1-\gamma)\lambda)}{1-\gamma} - \frac{\langle \nabla h((1-\gamma)\lambda), (1-\gamma)\lambda' - (1-\gamma)\lambda \rangle}{1-\gamma}\right]$$

Now, since $(1-\gamma)\lambda, (1-\gamma)\lambda' \in \Delta_{\mathcal{S}_+ \times \mathcal{A}}$ and $h$ is strongly convex on this set, we have

$$D_\Lambda(\lambda', \lambda) \geq \left(\frac{1}{1-\gamma}\right)^2 \frac{\|(1-\gamma)\lambda - (1-\gamma)\lambda'\|_1^2}{2} = \frac{\|\lambda - \lambda'\|_1^2}{2}.$$

Thus we have shown that $h_\gamma$ is 1-strongly convex on $\Lambda_\gamma$. By the properties of the negentropy function, we can verify that $h_\gamma(\lambda)$ is minimized for $\lambda_0 = \mathbf{1}_{\mathcal{A}}/A \oplus \gamma \mathbf{1}_{mA}/(1-\gamma)mA$, i.e. the uniform distribution over actions concatenated with the scaled uniform distribution over state-action pairs in $\mathcal{S}_* \times \mathcal{A}$ — this value is used as the initializer in Algorithm 1. Conversely, $h_\gamma(\lambda)$ is maximized for $\bar{\lambda} = e_a \oplus \gamma e_{s'a'}/(1-\gamma)$, i.e. when $\bar{\lambda}$ is concentrated on $(s_0, a)$ for some $a \in \mathcal{A}$ and some $(s', a') \in \mathcal{S}_* \times \mathcal{A}$. We can verify through a short calculation that

$$D_\Lambda(\bar{\lambda}, \lambda_0) = \frac{\ell}{(1-\gamma)^2}, \qquad \text{where } \ell := \log A + \gamma \log m$$

$$\Omega_\Lambda = \sqrt{2 D_\Lambda(\bar{\lambda}, \lambda_0)} = \frac{\sqrt{2\ell}}{1-\gamma}.$$

Finally, the proximal projection onto $\Lambda_\gamma$ with respect to $h_\gamma$ is

$$\Pi_\Lambda(\lambda, \rho) = \frac{\tilde{\lambda}_0}{\|\tilde{\lambda}_0\|_1} \oplus \frac{\gamma \tilde{\lambda}_*}{(1-\gamma)\|\tilde{\lambda}_*\|_1}, \qquad \text{where } \tilde{\lambda} := \exp(\log \lambda + \rho),$$

where $\tilde{\lambda}_{s_0} := [\tilde{\lambda}_{s_0 a}]_{a \in \mathcal{A}}$ and $\tilde{\lambda}_* := [\tilde{\lambda}_{sa}]_{s \in \mathcal{S}_*, a \in \mathcal{A}}$, so that $\tilde{\lambda} = \tilde{\lambda}_{s_0} \oplus \tilde{\lambda}_*$.

### A.4.4 The Composite Space

We will now gather together the preceding results and use them to construct a norm and distance-generating function on the composite optimization domain $Z = \Lambda_\gamma \times \mathcal{B} \subset \mathbb{R}^{(1+m)A} \oplus \mathbb{R}^d$. We closely follow the construction of Juditsky et al. [22, §4.2]. First, we define the squared norm:

$$\|\lambda \oplus \theta\|^2 := \Omega_\Lambda^{-2}\|\lambda\|_1^2 + \Omega_\mathcal{B}^{-2}\|\theta\|^2 = \left(\frac{1-\gamma}{\sqrt{2\ell}}\|\lambda\|_1\right)^2 + \left(\frac{1}{B}\|\theta\|\right)^2.$$

The corresponding squared dual norm is

$$\|\rho \oplus \xi\|_*^2 := \Omega_\Lambda^2\|\lambda\|_\infty^2 + \Omega_\mathcal{B}^2\|\theta\|_*^2 = \left(\frac{\sqrt{2\ell}}{1-\gamma}\|\lambda\|_\infty\right)^2 + (B\|\theta\|)^2.$$

Define the operator $F(\lambda, \theta) := -f_\lambda(\theta) \oplus f_\theta(\lambda)$. Its Lipschitz constant with respect to this norm is

$$\|F(\lambda, \theta)\|_* = \sqrt{\frac{2\ell\|f_\lambda(\theta)\|_\infty^2}{(1-\gamma)^2} + B^2\|f_\theta(\lambda)\|_*^2}$$

$$\leq \sqrt{\frac{8\ell B^2}{(1-\gamma)^2} + \left(\frac{2B}{1-\gamma}\right)^2}$$

$$= \frac{2B\sqrt{1+2\ell}}{1-\gamma} = C.$$

Similarly, define the estimator $\hat{F}(\lambda, \theta) := -\hat{f}_\lambda(\theta) \oplus \hat{f}_\theta(\lambda)$. Its variance enjoys

$$\mathbb{E}[\|\hat{F}(\lambda, \theta) - F(\lambda, \theta)\|_*^2] \leq C^2.$$

We do not repeat the calculation because it is identical to the previous one, since the variances of our estimators have the same bounds as their squared Lipschitz constants.

Finally, we construct the composite distance-generating function:

$$\omega_Z(\lambda, \theta) := \frac{(1-\gamma)^2 h_\gamma(\lambda)}{2\ell} + \frac{\|\theta\|^2}{2B^2}.$$

We can verify that this function is 1-strongly convex on $Z$ with respect to the norm defined above, and that the diameter of $Z$ under this function is $\Omega \le \sqrt{2}$.

*Proof of Lemma 11.* We apply the result of Juditsky et al. [22, Corollary 1] to our setting, where the Lipschitz constant of $F$ is $C$, the variance of $\hat{F}$ is $C^2$, and the diameter of $Z$ is $\Omega \le \sqrt{2}$. Then the result tells us that a suitable learning rate is $\eta = C^{-1}\sqrt{2/7T}$, as specified in Theorem 3, and the resulting bound on the expected duality gap after $T$ iterations is

$$\varepsilon_{\text{opt}} := \mathbb{E}[\delta_{\mathcal{B}}(\hat{\lambda}, \hat{\theta})] \le \frac{14C}{\sqrt{3T}}. \qquad \square$$

## Footnotes

[4]More generally, $\|\xi\|_* = \inf\{\|\eta\|_2 \mid \xi^\mathsf{T} = \eta^\mathsf{T}\Phi_*\}$, which is non-zero when $\xi \neq 0$ and $\Phi_*$ has full column rank.