[Reviews · NeurIPS 2020]

Review 1

Summary and Contributions: This paper considers the planning problem in discounted MDPs with linear function approximation. Under uniform approximation error (epsilon_approx) for only optimal value function (i.e. weak LFA), a (size m) set of core states, and generative oracles, the proposed algorithm needs poly(1/epsilon, m, A, 1/(1-gamma) queries from the generative oracle to achieve O(epsilon_approx/(1-gamma)^2 + epsilon) value loss. This work extends the previous work [26] by turning the error bound of relaxed ALP to an efficient planning algorithm. Specifically, this work follows a standard approach by proposing a stochastic saddle-point problem and then applies stochastic mirror-prox to solve it. The authors also improve the statistical and computational result in [26].

Strengths: Under the extra assumption (core states), this paper solves the efficient planning problem under weak LFA. This paper also improves value loss by 1/(1-gamma) the computational efficiency in [26].

Weaknesses: The algorithm requires additional core states assumption and this paper only studies the planning problem (i.e. requires generative oracle). The core states idea, the optimization problem, and the error bound are analyzed or hinted in [26]. This work extends on that and proposes a stochastic saddle-point problem and uses stochastic mirror-prox to solve it. This method seems to be a standard approach.

Correctness: I check the proof in high level due to the time limit. The results and proof seem to make sense to me. I can take a closer look if needed.

Clarity: This paper is clearly written and easy to follow. The pros and cons of the proposed approach is discussed in detail.

Relation to Prior Work: This paper has a thorough and clear discussion about the previous work with the contributions well-placed in the literature.

Reproducibility: Yes

Additional Feedback: After rebuttal: I read the author response and other reviews. Thanks the authors for addressing my concerns! It would be great to see more discussions in the next version. I think this paper has enough contribution and I will keep my original rating for acceptance. -------------------------------------------------------------------------------------------------------------------------------- People have been studying the weaker notion of LFA from different angles (see Table 1 in [13] for reference). This paper studies this problem under additional core states assumption. This paper is well written, and easy to follow. The main text is clean and the proof is deferred to the appendix. The authors also have an extensive discussion about the related work, which is quite useful. I have some detailed comments as below: - This paper mostly builds on [26]. The main idea and the error bound are discussed in [26] and this paper further proposes an efficient planning algorithm to solve the problem. In addition, this work improves the value loss by 1/(1-gamma) compared with [26]. I think the proposed approach is somewhat standard in optimization and is not very novel. - The efficient algorithm under weak LFA is an interesting and unsolved question. This paper proceeds in this direction. The proposed algorithm is computationally efficient and is suitable for stochastic environment without gap assumption. On the downside, the algorithm requires core states assumption and generative oracle. It seems that the generative oracle is required for the LP type of algorithm and the learning problem is beyond the scope of this paper. It would be interesting to see future work that further removes the core state assumption and solves the learning problem. - Currently the approximation error is defined in a uniform way. Do you think there are other meaningful ways to define the approximation error? I'm also wondering whether the approximation error can be defined in other norms. - As discussed around line 247, achieving O(epsilon_approx) error requires exp(H) queries while O(sqrt{d} H^2 epsilon_approx) optimal is possible in polynomial queries. It seems that relaxing to O(sqrt{d} epsilon_approx) is enough? Moreover, in Corollary 4, it is shown that the algorithm only needs poly(1/epsilon, m, A, 1/(1-gamma)) queries to get O(epsilon_approx/(1-gamma)^2 + epsilon). Does it mean that sqrt{d} blowup in the approximation error can be avoided (may be the price here is dependence on m, A and worse 1/(1-gamma)? I think such sqrt{d} has shown up in many related works that have DP flavor. Maybe it is avoided because of using LP? - Is it possible to extend the analysis in this paper to handle the case that the core states only approximately represent all the features? In other words, some features can lie outside the convex hull expanded by the core states. It is definitely possible to include those states in the core states, but I'm imagining the case that doing so may significantly increase the size of the core states and people prefer to use a small core states set.


Review 2

Summary and Contributions: This paper on the theory of MDP planning positively answers the open question of whether there exists a polynomial-time planner using a generative model of the MDP. The solution depends on two key assumptions: (1) having a known feature set, and mapping from states to that feature set, and (2) existence (and knowledge) of a small number of “core states”, such that the feature vector of every other state can be written as a positive linear combination of core features. Beyond the theoretical analysis, the paper provides an algorithm that achieves the result, and a thorough discussion of related work. No experimental results are provided.

Strengths: The paper identifies a very precise question, explains it clearly and concisely, and situates it well within the relevant literature. The core question is relevant to the MDP planning literature for large MDPs, it is based on a recent conjecture. The theoretical results and algorithmic techniques can be useful to solve further problems.

Weaknesses: Taken together, assumptions 1 and 2 seem quite strong, and hard to satisfy in practice. Another weakness of the paper is the lack of any empirical results, even simple demonstrations. Most of the claims and contributions of a theoretical nature, so it is not a fatal flaw. But it would have been nice in particular to see an empirical verification of what happens when the choice of feature space, or choice of core states deviates from the assumptions. I expect that satisfying both assumptions can be difficult in practice, so having some sense of robustness to misspecification would be interesting, even on simple domains.

Correctness: The results appear correct, though I did not carefully check the proofs.

Clarity: The paper is clear and well organized. The core technical section (on CoreStoMP) may be dense for the casual reader, but the language is precise, terms are defined and the authors take care in explaining several design choices and trade-offs in the algorithm and analysis.

Relation to Prior Work: The section on Related Work is particularly nice, not only does it cover relevant literature, but it carefully explains the salient differences from the current contributions. Several very recent works are discussed, but older references are also included for a more complete picture.

Reproducibility: Yes

Additional Feedback: - It would be interesting to comment further on what techniques might be used to discover or select the set of core states, and how to jointly discover the features space and the core states. - The style of the references is not consistent and should be cleaned up. Some use conference name abbreviation, others not, etc. Some ([8], [31]) don’t say where the work was published.


Review 3

Summary and Contributions: In this paper, the authors study the problem of designing an efficient RL algorithm when weak features are available. The authors propose the CoreStoMP algorithm, which guarantees near-optimal performance under two more assumptions (core states, simulator) with polynomial running time and the running time doesn't depend on the number of states. The CoreStoMP algorithm chooses an action by using Stochastic Mirror-Prox to solve the constructed CoreLP problem, and leveraging the fact that if at every state, the expected reward of the policy is near optimal, then the policy is near optimal.

Strengths: This paper answers an open problem by Du et al. in part. Although the problem is not completely solved, this proposed algorithm is definitely a step towards solving it.

Weaknesses: The assumption of Core States seems to be strong, which might prevent further application of the algorithm. For example, to my understanding, the algorithm can potentially work with continuous state space, but in that case, the number of core states might be exponential in d or even infinite.

Correctness: I don't check the math very carefully, but the theorem looks reasonable.

Clarity: The writing quality is good. Notations are well defined and the organization of the paper makes it easy to follow.

Relation to Prior Work: The authors discuss related work in Section 5 in details, which is quite useful to see how the work differs from prior work.

Reproducibility: Yes

Additional Feedback: Post Rebuttal: I've read the rebuttal. I didn't have many questions and would like to keep my score.

[Author Response · NeurIPS 2020]

We thank the reviewers for their feedback. We took great care to precisely state our research question and situate it amongst the prior work, and appreciate that the reviewers found this effort useful. We will definitely use the reviewers' feedback to improve the clarity further. We will now try to address some of the specific concerns, and add these clarifications to the paper.

**Core states:** We agree with all the reviewers that the assumption of core states is quite strong, and we are highly interested in generalizing and weakening this condition. In particular, we would like to: (1) "localize" the condition, requiring only core states that cover the successor states of the current planning state, rather than the entire state space; (2) weaken the condition by considering "approximate" core states, whose convex hull covers "most" of the probable next states. Then one would expect a similar approximation error, with additional dependence on the probability of visiting states whose features lie outside the core set convex hull.

We hope that these questions will be the subject of future work. Nevertheless, we feel that the contribution is useful in its present form. In particular, as we discuss in the Related Work section, it is still not known how to avoid exponential blow-up of computation in the planning horizon without such strong assumptions. Furthermore, as we also discuss in the paper, unlike other assumptions that have been made in prior work (e.g. linear transition models, low Bellman rank, etc.) ours is a purely geometric condition on the feature representation, *not* a restriction on the transition model of the MDP. We foresee two basic approaches to satisfying this assumption in practice: (1) adapt exploration methods to discover core states; (2) in practice, large scale reinforcement learning usually requires representation learning — the learned representations could be made to satisfy the core state assumption *by design*. This option is not available when restrictions are placed on the MDP itself, rather than the feature representation. Both of these approaches are quite intriguing, require careful design and analysis, and will be the subject of future work. We believe that our results in this paper lay the foundations for more exhaustive research in this direction, and will discuss these possibilities in the paper.

**Novelty of the optimization approach (Reviewer 1):** The linear programming approach to control in RL has, of course, a long history. It is also quite standard in the optimization literature to use saddle point methods to solve linear programs. However, it remains a tricky problem to actually *use* the solutions of an approximate saddle-point solver for planning and control in RL. We hint at some of the challenges in the main paper, and will definitely add more discussion about this along with our techniques. This algorithmic issue is important and not addressed by Lakshminarayanan et al.

The solutions produced by approximate saddle point algorithms are not just suboptimal — they violate the constraints of the original linear program. This is especially problematic with function approximation, where the interpretation of the dual LP solution as an occupancy measure is no longer meaningful. Bas-Serrano and Neu [arxiv:1909.10904] have recently pointed out that, without an additional (strong) "coherence" assumption, the policies extracted from approximate saddle point solutions can be arbitrarily suboptimal. They also argue that other prior works suffer from this issue. The problem is also difficult because saddle solvers needed bounded constraint sets. We will add this discussion to the paper.

One of our contributions in this paper (which we will expand upon) is to show that the core state assumption not only bounds the approximation error of the LP, but also solves the algorithmic problem of using approximate saddle point solutions to identify good actions. We hope that our techniques will be useful for future algorithm designers.

**Achievable error (Reviewer 1):** It is an interesting question whether polynomial-time planning under function approximation is possible without the extra $H^2$ factor in the error bound. We believe that Theorem 1 of Tsitsiklis and Van Roy [doi:10.1007/BF00114724] shows that the $H^2$ factor is unavoidable, and will expand on this in the paper.

In our work the expected $\sqrt{d}$ factor is replaced by $\sqrt{m}$, where $m$ is the number of core states. Importantly, if the feature matrix is full-rank then $m \geq d$, so we are not avoiding the dimension-dependence. However, we believe that improving the optimization method can improve the dependence on $\sqrt{m}$ to $\sqrt{d}$ and hope to achieve this in the future. The number of core states $m$ is related to the geometry of the feature-set, regardless of its cardinality; $m$ can be finite for infinite state spaces (**Reviewer 4**).

**Uniform approximation (Reviewer 1):** $\varepsilon_{\mathrm{approx}}$ is an $\infty$-norm error bound and thus measures approximation error uniformly over all states. Prior works have considered weighted norms (e.g. Lakshminarayanan et al.) and we believe those results can be transferred to our work, at the price of additional assumptions (the Lyapunov stability condition). We opted not to make this extension to keep the paper focused, but will note the possibility in the discussion.

**Experimental results (Reviewer 2):** Although the focus of this work was theoretical, we will endeavour to provide some representative numerical results, possibly in the Supplementary Material. We will also refer readers to the simulations of Lakshminarayanan et al., which address the quality of the relaxed ALP solutions, if not the efficiency of the optimization algorithms presented in this paper.

[Meta-Review · NeurIPS 2020]

All reviewers agree that the paper makes a nice contribution to planning with function approximation. In particular, the paper considers an important open problem, and while the problem is solved by making a few assumptions (mostly notably the core states), the results have made significant progress on the important problem. The reviewers also appreciate the use of precise language and careful description of related work. Among the remaining concerns, R2 wants to see some evidence of robustness against the failure of the "core state" assumption. While performing empirical experiments may not fit the theoretical nature of the paper, the authors can consider a theoretical justification: namely, define a notion of error that measures how much the core-states assumption is violated, and show how such an error manifest itself in the final guarantee. How much of a blow-up (hopefully polynomial) will we get? Sketching out such an analysis (possibly in the appendix) would help answer the robustness question.